# Regional irreversibility of mean and extreme surface air temperature and precipitation in CMIP6 overshoot scenarios associated with interhemispheric temperature asymmetries

Pedro José Roldán-Gómez[1], Paolo De Luca[1], Raffaele Bernardello[1], and Markus G. Donat[1,2]

[1]Barcelona Supercomputing Center, Earth Sciences Department, Barcelona, Spain
[2]Institució Catalana de Recerca i Estudis Avançats (ICREA), Barcelona, Spain

**Correspondence:** Pedro José Roldán-Gómez (pedro.roldan@bsc.es)

**Abstract.**

Overshoot scenarios, in which the forcing reaches a peak before starting to decline, show non-symmetric changes during the $CO_2$ increasing and decreasing phases, producing persistent changes on climate. Irreversibility mechanisms, associated among others with lagged responses of climate components, changes in ocean circulation and heat transport and changes in the ice cover, bring hysteresis to the climate system. This work analyzes simulations from the Coupled Model Intercomparison Project Phase 6 (CMIP6) to explore the relevance of these mechanisms in overshoot scenarios with different forcing conditions (SSP5-3.4OS and SSP1-1.9) and the impact on regional climates, with a particular focus on the degree to which changes in regional extremes are reversible. These analyses show that in scenarios with strong forcing changes like SSP5-3.4OS, the post-overshoot state is characterised by a temperature asymmetry between Northern and Southern Hemisphere, associated with shifts of the Intertropical Convergence Zone (ITCZ). In scenarios with lower forcing changes like SSP1-1.9, this hemispheric asymmetry is more limited and temperature changes in polar areas are more prominent. These large scale changes have an impact on regional climates, such as for temperature extremes in extratropical regions and for precipitation extremes in tropical regions around the ITCZ. Differences between pre- and post-overshoot states may be associated with persistent changes in the heat transport and with a different thermal inertia depending on the region, leading regionally to a different timing of the temperature maximum. Other factors like changes in aerosol emissions and ice melting may be also important, particularly for polar areas. Results show that irreversibility of temperature and precipitation extremes is mainly caused by the transitions around the global temperature maximum, when a decoupling between regional extremes and global temperature generates persistent changes at regional level.

## 1 Introduction

The Paris Agreement of 2015 included an objective to limit the increase in the global average temperature to well below 2°C above pre-industrial levels and to pursue efforts to limit it to 1.5°C (United Nations / Framework Convention on Climate Change, 2015). However, considering the delay of effective and consequent mitigation measures (IPCC, 2022), there is an increasing probability to exceed these temperature targets (Raftery et al., 2017). In this scenario, global average temper-

ature might overshoot the targets of the Paris Agreement and net-negative emissions would be needed to reduce global $CO_2$
concentrations and bring temperatures back to a level consistent with the targets (Gasser et al., 2015).

To address questions related to such delayed climate action, there is an increasing interest in scenarios with forcing pathways that reach a peak before starting a forcing decline, also known as overshoot scenarios. The Coupled Model Intercomparison Project Phase 6 (CMIP6; Eyring et al., 2016) included two scenarios with these characteristics in the ScenarioMIP (O'Neill et al., 2016): SSP5-3.4OS, which follows the unmitigated scenario SSP5-8.5 up to 2040 and starts an aggressive mitigation
afterwards; and SSP1-1.9, that includes mitigation actions to meet the 1.5°C target from the Paris Agreement (Tebaldi et al., 2021). These scenarios allow investigating potentially irreversible changes in the climate system as a result of a cycle of increasing and decreasing forcing (IPCC, 2022), considering that even in case global temperatures revert, the impact on regional climates, and in particular on regional temperature, precipitation and climate extremes, may remain for decades (Pfleiderer et al., 2024).

The analysis of irreversibility mechanisms has been mostly based on idealized $CO_2$ ramp-up and ramp-down experiments: Zickfeld et al. (2016) show that the proportionality between global mean temperature and $CO_2$ emissions does not persist during periods of net negative $CO_2$ emissions, mostly due to a different behavior of ocean and land, while Boucher et al. (2012) show that certain climate components like clouds and ocean stratification respond with a lag with respect to temperatures, generating a hysteresis behavior. Hysteresis, understood as the dependence of the climate system not only on the current $CO_2$ concentration
but on the $CO_2$ pathway, is also found in carbon sinks (Jeltsch-Thömmes et al., 2020), surface air temperatures (Jones et al., 2016), melting of ice sheets (Bochow et al., 2023), and ocean carbon cycle feedbacks (Schwinger and Tjiputra, 2018), with an impact in the ocean circulation and sea level changes (Palter et al., 2018). Hysteresis also appear in the location of the Intertropical Convergence Zone (ITCZ; Kug et al., 2022), changing minimally during the ramp-up period but experiencing a relevant southward displacement during the ramp-down. Kug et al. (2022) associate this hysteresis in the position of the ITCZ
with a delayed energy exchange between the tropics and extratropics, linked to changes in the Atlantic Meridional Overturning Circulation (AMOC) and in the temperature of the Southern Ocean.

In general, changes in the position of the ITCZ can be explained by temperature asymmetries and changes in the meridional heat transport (Donohoe et al., 2013). These changes are particularly relevant over oceans (Chiang and Bitz, 2005), with a northward displacement of the Pacific ITCZ in response to the cooling of the eastern Pacific (Takahashi and Battisti, 2006)
or with a southward shift of the Atlantic ITCZ linked to the cooling of the northern Atlantic (Vellinga and Wood, 2002). Temperature asymmetries behind these changes have been associated to changes in the AMOC (Moreno-Chamarro et al., 2020), changes in the ice cover (Chiang and Bitz, 2005), alterations of the Thermohaline Circulation (THC; Zhang and Delworth, 2005), and the asymmetry introduced by orography (Takahashi and Battisti, 2006). In larger timescales, changes in the position of the ITCZ have been found in simulations of the Last Glacial Maximum (Chiang et al., 2003), linked to an asymmetric
cooling between the Northern Hemisphere (NH) and the Southern Hemisphere (SH) generated by a change in the amount of polar sea ice, variations in surface albedo, and changes in the THC (Lohmann, 2003), as well as during the Last Millennium (Roldán-Gómez et al., 2022), induced both by external forcing and internal variability.

**Table 1.** Climate models analyzed, available simulations for the experiments SSP5-3.4OS and SSP1-1.9 considered in this work, number of latitude and longitude levels of each model and associated references. For the SSP5-3.4OS, the number of simulations covering the extended period (up to 2300) is also included in the column (EXT).

| Model | SSP5-3.4OS | (EXT) | SSP1-1.9 | N Lon | N Lat | References |
|---|---|---|---|---|---|---|
| ACCESS-CM2 | 1 (r1i1p1f1) | 0 | 0 | 192 | 144 | Ziehn et al. (2021) |
| CanESM5 | 5 (r[1-5]i1p1f1) | 1 (r1i1p1f1) | 50 (r[1-25]i1p[1-2]f1) | 128 | 64 | Swart et al. (2019a, b) |
| CMCC-ESM2 | 1 (r1i1p1f1) | 0 | 0 | 288 | 192 | Lovato et al. (2021) |
| CNRM-ESM2-1 | 1 (r1i1p1f2) | 1 (r1i1p1f2) | 1 (r1i1p1f2) | 256 | 128 | Voldoire (2019a, b) |
| EC-Earth3 | 0 | 0 | 6 (r[1-4]i1p1f1) | 512 | 256 | Döscher et al. (2022); EC-Earth-Consortium (2019a, b, c) |
| FGOALS-g3 | 1 (r1i1p1f1) | 0 | 1 (r1i1p1f1) | 180 | 80 | Li (2019, 2020) |
| GFDL-ESM4 | 0 | 0 | 1 (r1i1p1f1) | 288 | 180 | John et al. (2018) |
| IPSL-CM6A-LR | 1 (r1i1p1f1) | 1 (r1i1p1f1) | 6 (r[1-4,6,14]i1p1f1) | 144 | 143 | Boucher et al. (2019a, b) |
| MIROC6 | 0 | 0 | 50 (r[1-50]i1p1f1) | 256 | 128 | Shiogama et al. (2019) |
| MIROC-ES2L | 0 | 0 | 10 (r[1-10]i1p1f2) | 128 | 64 | Tachiiri et al. (2019) |
| MPI-ESM1-2-LR | 0 | 0 | 30 (r[1-30]i1p1f1) | 192 | 96 | Schupfner et al. (2021) |
| MRI-ESM2-0 | 1 (r1i1p1f1) | 1 (r1i1p1f1) | 5 (r[1-5]i1p1f1) | 320 | 160 | Yukimoto et al. (2019a, b) |
| UKESM1-0-LL | 5 (r[1-4,8]i1p1f2) | 0 | 5 (r[1-4,8]i1p1f2) | 192 | 144 | Good et al. (2019a, b) |

Despite their characterization with idealized experiments, the relevance of these hysteresis mechanisms in more plausible scenarios is not evident. Scenarios with net zero $CO_2$ emissions show temperature asymmetries between continental areas and the Southern Ocean, with changes that persist well beyond the stabilization (King et al., 2024), as well as a larger incidence of warm extremes in regions of the south and cold extremes in regions of the north (Cassidy et al., 2024). For the SSP5-3.4OS scenario, Melnikova et al. (2021) found carbon cycle feedbacks over land and ocean, and Pfleiderer et al. (2024) showed regional changes in areas of Western and Central Africa consistent with ITCZ shifts. However, the analyses from Walton and Huntingford (2024) do not show a relevant hysteresis on regional precipitation. This shows the need of further analyzing the role of hysteresis mechanisms in shaping regional climates in overshoot scenarios, including both temperature and precipitation extremes.

Both observational (Donat et al., 2013; Dunn et al., 2020) and simulated data (Sillmann et al., 2013) show relevant changes in climate extremes in a context of global warming, with human activities contributing to changes of temperature (Kim et al., 2016) and precipitation extremes (Zhang et al., 2013; Min et al., 2011). Seneviratne et al. (2016) showed that the evolution of regional temperature and precipitation extremes is mostly proportional to the cumulative $CO_2$ emissions and to the increase of global temperatures, with a different sensitivity depending on the region. However, the presence of this proportionality in a context of decreasing $CO_2$ concentrations and decreasing global temperatures has not been analyzed.

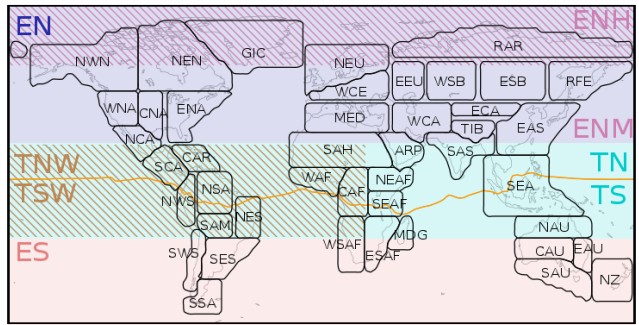

**Figure 1.** Regions considered for the analysis of extremes, including IPCC climate reference regions, as defined in Iturbide et al. (2020), extratropical areas of the NH (EN; 23° N - 90° N), high-latitude extratropical areas of the NH (ENH; 60° N - 90° N), mid-latitude extratropical areas of the NH (ENH; 23° N - 60° N), extratropical areas of the SH (ES; 90° S - 23° S), tropical areas north of the 2020-2039 ITCZ (TN; ITCZ - 23° N), tropical areas south of the 2020-2039 ITCZ (TS; 23° S - ITCZ), tropical areas north of the 2020-2039 Atlantic and eastern Pacific ITCZ (TNW; ITCZ - 23° N; 180° W - 25° E), and tropical areas south of the 2020-2039 Atlantic and eastern Pacific ITCZ (TSW; 23° S - ITCZ; 180° W - 25° E).

In line with Pfleiderer et al. (2024), this work analyzes overshoot scenarios from CMIP6 (SSP5-3.4OS and SSP1-1.9) to investigate how global changes in temperature and precipitation during the overshoot are associated with regional irreversibility. Irreversibility is understood as a post-overshoot state different from the pre-overshoot state, considering pre-overshoot and post-overshoot states with the same $CO_2$ concentration and with the same global temperature. This includes then continued, partially reversed and overcompensated behaviors, as described in Pfleiderer et al. (2024). Contrary to Pfleiderer et al. (2024), who focuses on the regional reversibility up to 2100, our work includes a detailed characterisation of the stabilisation period, including also simulations of SSP5-3.4OS extending up to 2300. The analyses go also deeper into the mechanisms explaining the different regional behaviors, with an evaluation of the changes in the position of the ITCZ as a result of persistent temperature asymmetries. These analyses, including mean and extreme climates, allow not only for identification of those regions more impacted by irreversibility, but also of the mechanisms explaining different regional behaviors.

## 2 Methods

The analyses have been focused on the overshoot scenarios from CMIP6 (SSP5-3.4OS and SSP1-1.9). For that, the simulations in Table 1 have been considered. For some of the models, several simulations with the same forcing specifications and different initial conditions are considered. It should be noted that for SSP1-1.9 all the simulations are run up to 2100, while for SSP5-3.4OS some extended simulations, expanding up to 2300, are also available. These extended simulations have been considered for a better characterization of the state after stabilization. Results are presented both for the ensemble of all simulations (ALL)

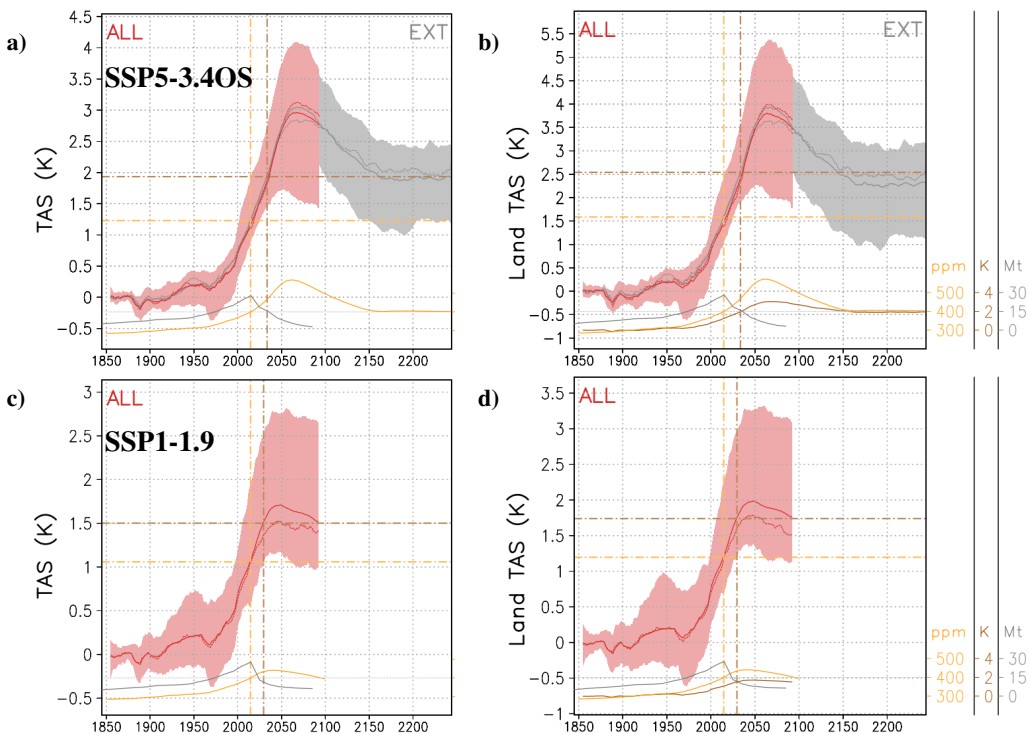

**Figure 2. (a,c)** Global and **(b,d)** land-only average of surface air temperature (TAS) anomaly with respect to 1861-1880 obtained from the CMIP6 simulations of experiments **(a,b)** SSP5-3.4OS and **(c,d)** SSP1-1.9, considering the average (solid line) and the median (dashed line) of all the models (ALL) and, for SSP5-3.4OS, the ensemble of simulations covering up to 2300 (EXT). The dispersion of individual simulations within each ensemble is included with a shading. Yellow, gray and brown curves in the lower part of each panel respectively show the $CO_2$ concentration from Meinshausen et al. (2020), the anthropogenic aerosol emissions (BC and OC) from Feng et al. (2020), and the global temperature obtained with the ALL ensemble for SSP1-1.9 and the EXT ensemble for SSP5-3.4OS. The vertical lines show the year before the overshoot with the same $CO_2$ concentration and global temperature as at the end of the run (2100 for the ALL ensemble of SSP1-1.9 and 2300 for the EXT ensemble of SSP5-3.4OS), while the horizontal lines represent the value of temperature in the ALL ensemble for SSP1-1.9 and in the EXT ensemble for SSP5-3.4OS for those years.

and, for the case of SSP5-3.4OS, for the ensemble of extended simulations (EXT). As shown in Table 1, for the case of SSP5-3.4OS the ALL ensemble is based on eight models and 16 simulations, while the EXT ensemble is only based on four models, with only one simulation per model. To validate the representativity of the EXT ensemble with respect to the ALL ensemble, the results up to 2100 have been obtained with both the ALL and EXT ensembles.

As shown in Table 1, each model has a different resolution, ranging from 128 to 512 longitude levels and from 64 to 256 latitude levels. To allow for combined analyses, all the simulations have been interpolated to a common grid resolution of 2.8125º x 2.8125º, the coarsest among the analyzed climate models. The ensemble average has been computed by averaging all the simulations of each model to obtain a per-model average in a first step and by averaging all the models in a second

step. The use of ensemble averages allows for a synthetic view of the results, but it may be not meaningful in case of large discrepancies across the contributing models, in particular in terms of global temperature trajectories during the overshoot and in terms of regional hysteresis. Other metrics like the ensemble median would be more robust to these effects, but they may be impacted by internal variability of individual simulations. To confirm that the ensemble average is not biased by any particular model, the ensemble median has also been computed and compared with the ensemble average. To analyze the dispersion among the models and within a single model, time series for the ensemble of each individual model providing several simulations (CanESM5 and UKESM1-0-LL for SSP5-3.4OS and CanESM5, EC-Earth3, IPSL-CM6A-LR, MIROC6, MIROC-ES2L, MPI-ESM1-2-LR, MRI-ESM2-0, and UKESM1-0-LL for SSP1-1.9) and examples of spatial patterns for some individual models (CNRM-ESM2-1 and MRI-ESM2-0 for SSP5-3.4OS, and CanESM5 and IPSL-CM6A-LR for SSP1-1.9) are also presented in Appendix A.

Analyses have been based on temperature and precipitation annual and seasonal averages, as well as on extreme indices, including the warmest day (TXx) and the coldest night (TNn) of the year, the percentage of time when the daily maximum temperature is above the 90th percentile (TX90p) and when the daily minimum temperature is below the 10th percentile (TN10p), and the annual maximum consecutive 5 day (Rx5day) and 1 day (Rx1day) precipitation total (Zhang et al., 2011). This set of indices allows for a characterization of both precipitation (Rx5day and Rx1day) and temperature extremes, including both warm (TXx and TX90p) and cold (TNn and TN10p) extremes, and considering both the absolute value (TXx and TNn) and the distribution (TX90p and TN10p). To remove short-term variability, analyses have been based on comparisons of 20 year periods and temporal evolutions filtered with a 10 year moving average.

The situation after stabilization has been compared with the situation before the overshoot with the same $CO_2$ concentration, as provided by Meinshausen et al. (2020), reached in 2015 both for SSP5-3.4OS and SSP1-1.9. It has been also compared with the situation with the same global temperature, reached in 2034 for SSP5-3.4OS and in 2030 for SSP1-1.9. Considering these dates and to use a reference period large enough to focus on the long-term variability, the period from 2020 to 2039 has been considered as pre-overshoot reference period for most of the analyses. To confirm the suitability of this reference period, results obtained with alternative pre-overshoot periods from 2010 to 2029 and from 2030 to 2049 have been also included in Appendix B. Even if changes in $CO_2$ concentration are the main contribution to the change of radiative forcing during the overshoot, changes in aerosols may also play a relevant role, particularly over the Arctic (England et al., 2021; Ren et al., 2020; DeRepentigny et al., 2022). For this reason, the emissions of aerosols for SSP5-3.4OS and SSP1-1.9 have been assessed, as provided by Feng et al. (2020).

The regional climate conditions have been analyzed with the averages for extratropical areas of the NH and SH, extratropical mid and high latitudes, and tropical areas to the north and to the south of the ITCZ, as well as by considering the updated IPCC climate reference regions from Iturbide et al. (2020) (Fig. 1). The ITCZ has been characterized with the precipitation centroid of the area between 20° S and 20° N, except for the Pacific basin, in which the southern branch of the ITCZ (Tian and Dong, 2020) has been removed by limiting the computation to the area between 0° and 20° N. To analyze changes in the position of the ITCZ, a comparison between the precipitation centroid before and after the overshoot has been performed.

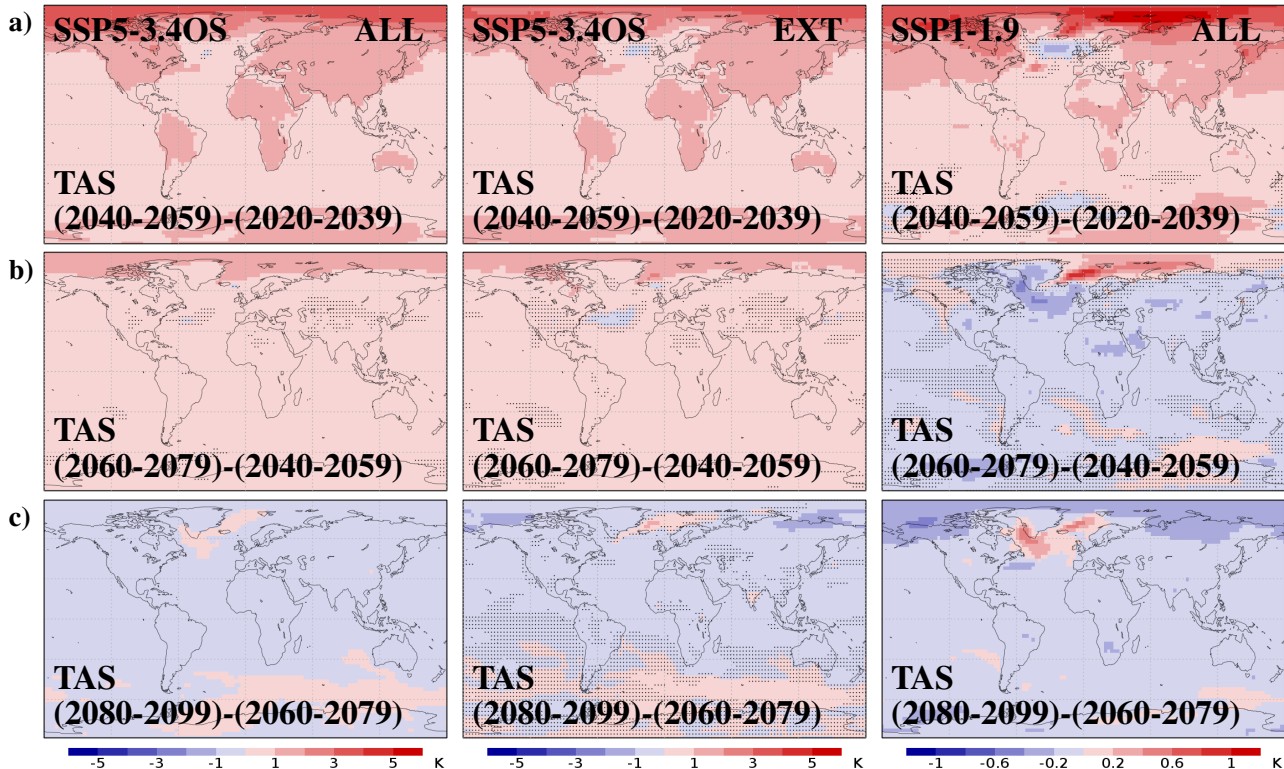

**Figure 3.** Difference between the ensemble mean, temporal average values of surface air temperature (TAS) for the periods **(a)** 2040-2059 and 2020-2039, **(b)** 2060-2079 and 2040-2059, and **(c)** 2080-2099 and 2060-2079, obtained with the **(left)** ALL ensemble and the **(center)** EXT ensemble of SSP5-3.4OS simulations, and with the **(right)** ALL ensemble of SSP1-1.9 simulations. Stippling indicates locations where the differences are not significant (t-test with p<0.05).

## 3  Results

### 3.1  Changes in mean surface air temperature

Figure 2a,c shows the global average of temperature for the experiments SSP5-3.4OS and SSP1-1.9, including both ensemble average and ensemble median. For both experiments, the global average reaches a maximum (in 2068 and 2050 respectively) and start to decrease afterwards. For the case of SSP5-3.4OS (Fig. 2a), the global average of the EXT ensemble, containing simulations extending up to 2300, shows a stabilization before the end of the scenario. For the SSP1-1.9 experiment (Fig. 2c) and the ALL ensemble of the SSP5-3.4OS (Fig. 2a), for which only simulations up to 2100 are available, the global average of temperature is still decreasing by the end of the simulations. The average of temperature anomalies only for land areas (Fig. 2b,d) is in general higher than the global average, consistent with a larger sensitivity of continental areas to changes in the forcing (Bindoff et al., 2013), and while the global average for the EXT ensemble of SSP5-3.4OS stabilizes to the same value

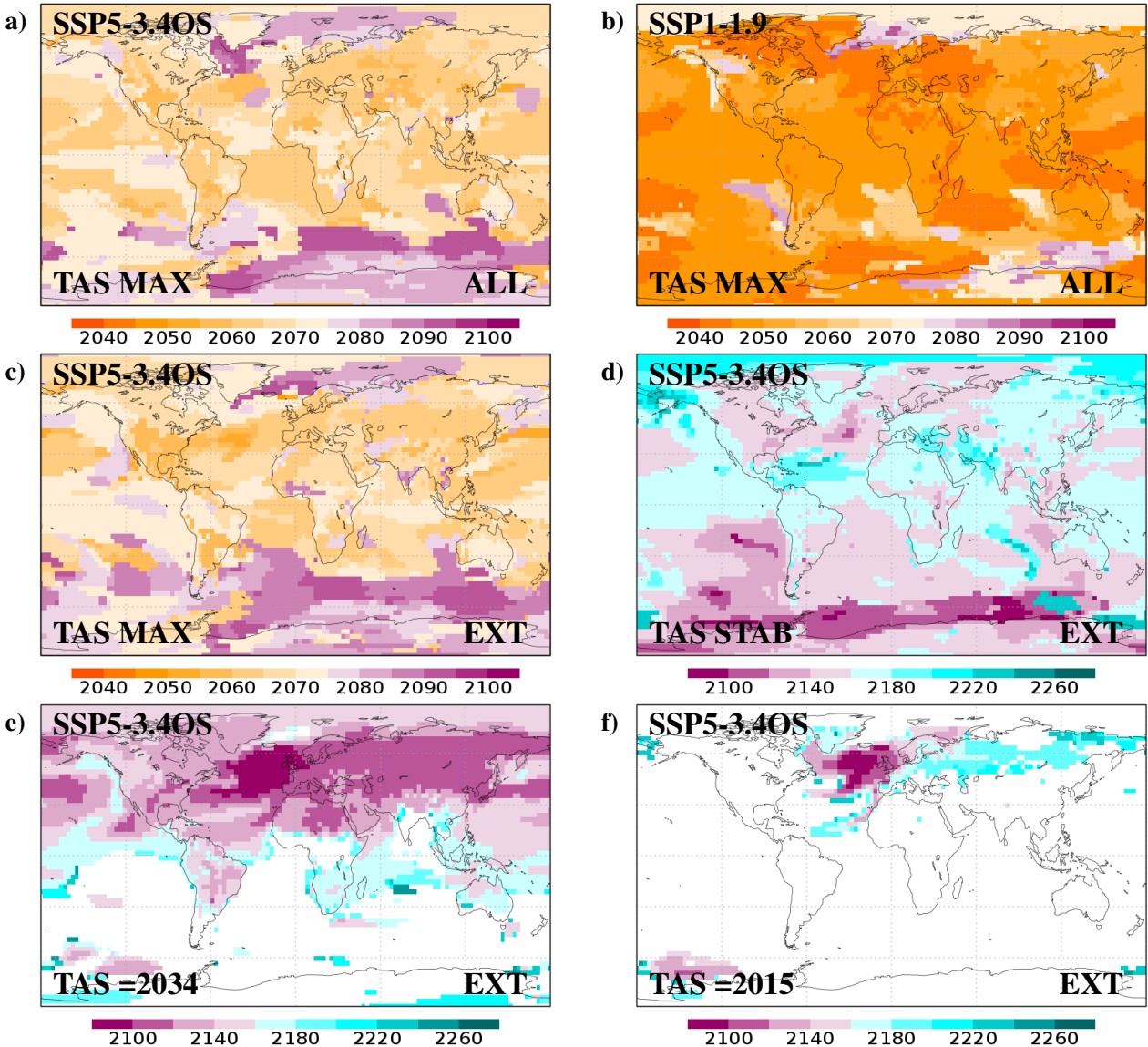

**Figure 4. (a)** Year of maximum surface air temperature (TAS) for the ALL ensemble of SSP5-3.4OS. **(b)** Year of maximum surface air temperature (TAS) for the ALL ensemble of SSP1-1.9. **(c)** Year of maximum surface air temperature (TAS) for the EXT ensemble of SSP5-3.4OS. **(d)** Year of stabilization of surface air temperature (TAS) for the EXT ensemble of SSP5-3.4OS, obtained as the year after the maximum in which temperature reaches the same value as in the period 2290-2300. **(e,f)** Year after the maximum in which surface air temperature (TAS) for the EXT ensemble of SSP5-3.4OS reaches the same value as in **(e)** 2034 (year before the overshoot corresponding to the same global temperature as in 2300) and **(f)** 2015 (year before the overshoot corresponding to the same $CO_2$ concentration as in 2300). Blank grid points indicate locations where the value is not reached before 2300.

as in 2034 (brown line in Fig. 2a,b), the average for land areas stabilizes to a lower value (Fig. 2b), suggesting a different behavior during the overshoot for oceanic and continental regions.

When analyzing the spatial patterns before and after the maximum of SSP5-3.4OS (Fig. 3), a first period with a strong increase of temperatures over continental and polar regions and a moderate increase over ocean is found (Fig. 3a). During this period, most regions show an increase of temperature, except for certain areas of the northern Atlantic, impacted by melting, changes in ocean heat transport and cloud feedbacks (Keil et al., 2020). The warming is more limited starting from 2060 (Fig. 3b), showing an impact of the mitigation of $CO_2$ emissions considered in the SSP5-3.4OS experiment. During the last 20 years of the century (Fig. 3c), temperatures start to decrease for most continental and tropical oceanic areas, while they continue to increase for some areas of the Southern Ocean and the northern Atlantic, potentially explained by ice melting and the inertia of ocean during the warming and cooling phases. Despite some minor differences, like a less widespread cooling between 2060 and 2099 over the SH and a more widespread cooling between 2040 and 2079 in areas of the northern Atlantic, the EXT ensemble generally shows a similar behavior to that of the ALL ensemble, both in terms of global averages (Fig. 2a,b) and spatial patterns (Fig. 3), showing that even if based on a limited number of simulations the EXT ensemble provides robust results.

For the case of SSP1-1.9, the lower forcing changes are associated with a more limited increase of temperatures (Fig. 3). The period from 2040 to 2059 (Fig. 3a) shows a relevant increase in temperatures with respect to the previous 20 years for most continental areas, while a decrease is found in some particular oceanic areas of the northern Atlantic and the Southern Ocean. Starting from 2060 (Fig. 3b), a decrease of temperatures is found for most regions, with the exception of polar areas of the NH, some continental areas like northwestern North America, and large areas of the Southern Ocean. The pattern of increasing and decreasing temperatures obtained with SSP1-1.9 resembles that of SSP5-3.4OS (Fig. 3b), in particular for the opposition between cooling in continental areas of the NH and persistent warming in areas of the Southern Ocean. However, SSP1-1.9 shows a persistent warming in most polar areas of the NH and cooling over the western Southern Ocean. This may be linked to the fact that even if $CO_2$ concentration strongly differs from SSP1-1.9 to SSP5-3.4OS, the anthropogenic aerosol emissions, more relevant in polar areas (England et al., 2021), are similar for both experiments (Fig. 2). The timing for both scenarios also differs, with most regions starting the decrease of temperatures before 2060 for SSP1-1.9 (Fig. 3b) and before 2080 for SSP5-3.4OS (Fig. 3c). This is also evident in the date of the maximum, which is reached for most regions before 2050 in the SSP1-1.9 experiment (Fig. 4b), and between 2060 and 2070 in the SSP5-3.4OS experiment (Fig. 4a). For certain areas of the Southern Ocean and polar areas of the NH the maximum is delayed, after 2080 in SSP1-1.9 and even after 2090 in SSP5-3.4OS.

As shown in Fig. 4, areas with a delayed maximum temperature (Fig. 4a) are not necessarily those with the latest stabilization (Fig. 4d). While the polar and oceanic areas tend to reach the maximum later, the tropical and continental areas are those showing the longest stabilization, with certain areas in the Caribbean, tropical Atlantic, eastern Mediterranean and the Indian basin not stabilizing before 2200 for the SSP5-3.4OS experiment (Fig. 4d). This may be linked to the presence of long-term mechanisms explaining changes in the tropics, like persistent alterations in the position of the ITCZ (Kug et al., 2022). Despite the fact that the global average of temperature reaches the same value as in 2034 after stabilization (Fig. 2a), this is mainly

limited to the NH and some tropical areas of the SH, with most of the SH stabilizing to higher temperatures (Fig. 4e). The temperatures of 2015, year with the same $CO_2$ concentration before the overshoot, are only recovered before 2100 in the northern Atlantic and before 2300 in some continental areas of Europe and central Asia (Fig. 4f).

The results from Fig. 4 indicate that after the overshoot the climate stabilizes to a situation that differs from that of before. As shown in Fig. 5a, for SSP5-3.4OS the situation after stabilization is characterized by a colder NH and a warmer SH compared to the pre-overshoot climate, with the largest negative and positive differences obtained for the highest latitudes. For SSP1-1.9 there is no evident temperature asymmetry between NH and SH by the end of the 21st century (Fig. 6a). Instead, higher temperatures are found on polar areas of the NH and in the eastern Southern Ocean. On the contrary, lower temperatures are found around West Antarctica, known to be impacted by ice melting even under low forcing conditions (Naughten et al., 2023). The forcing conditions of SSP1-1.9 are then characterized by an opposition between high and mid latitudes rather than an opposition between NH and SH, potentially due to a delayed recovery of sea ice (Bauer et al., 2023) and a larger role of anthropogenic aerosol emissions (Fig. 2). The changes for SSP1-1.9 are in general more limited than those observed for SSP5-3.4OS (Fig. 5a), mostly due to a much weaker overshoot, but also to the fact that for SSP1-1.9 there are no simulations extending up to 2300 and the stabilization is not fully reached by 2100.

## 3.2   Changes in mean precipitation

The temperature asymmetries of SSP5-3.4OS explain differences in the spatial distribution of precipitation, as shown in Fig. 5b. Precipitation after stabilization tends to be higher for areas south of the annual mean ITCZ and lower for areas to the north, indicating a southward shift of the ITCZ. This shift reaches at certain longitudes 1º for the annual ITCZ, and more than 2º when considering the ITCZ for December-January-February (DJF; Fig. 5d) and June-July-August (JJA; Fig. 5e). The southward shift is particularly strong over the Atlantic and the eastern Pacific, while the Indian and western Pacific present more limited shifts, and even northward shifts at certain longitudes. In the Atlantic basin, the changes in the ITCZ position can be associated with changes in the ocean heat transport (Fig. 5c), linked to a decline of the AMOC.

For SSP1-1.9, the limited temperature asymmetry between NH and SH explains more limited ITCZ shifts (Fig. 6b,d,e), only reaching 0.2º for some areas of the Atlantic and Indian basin. For this scenario, the ITCZ shifts are to the south in the Pacific and Atlantic basin and to the north in the Indian basin (Fig. 6b).

## 3.3   Changes in extreme surface air temperature

Considering that a moderate impact on global climate may have a strong impact on regional extremes (Seneviratne et al., 2016), it could be expected that the persistent large-scale temperature changes found in SSP5-3.4OS and SSP1-1.9 scenarios significantly alter the extremes in certain regions.

Changes in the extreme indices TXx, TNn, TX90p, and TN10p between the situation before and after the overshoot of SSP5-3.4OS are shown in Fig. 7. Temperature extremes show an opposite behavior between regions in the NH and SH (Fig. 7a-d), consistent with the opposite behavior of average temperatures shown in Fig. 5a. Increase in intensity and frequency of the warmest temperatures is particularly relevant in areas of Western and Central Africa and India (WAF, CAF and SAS;

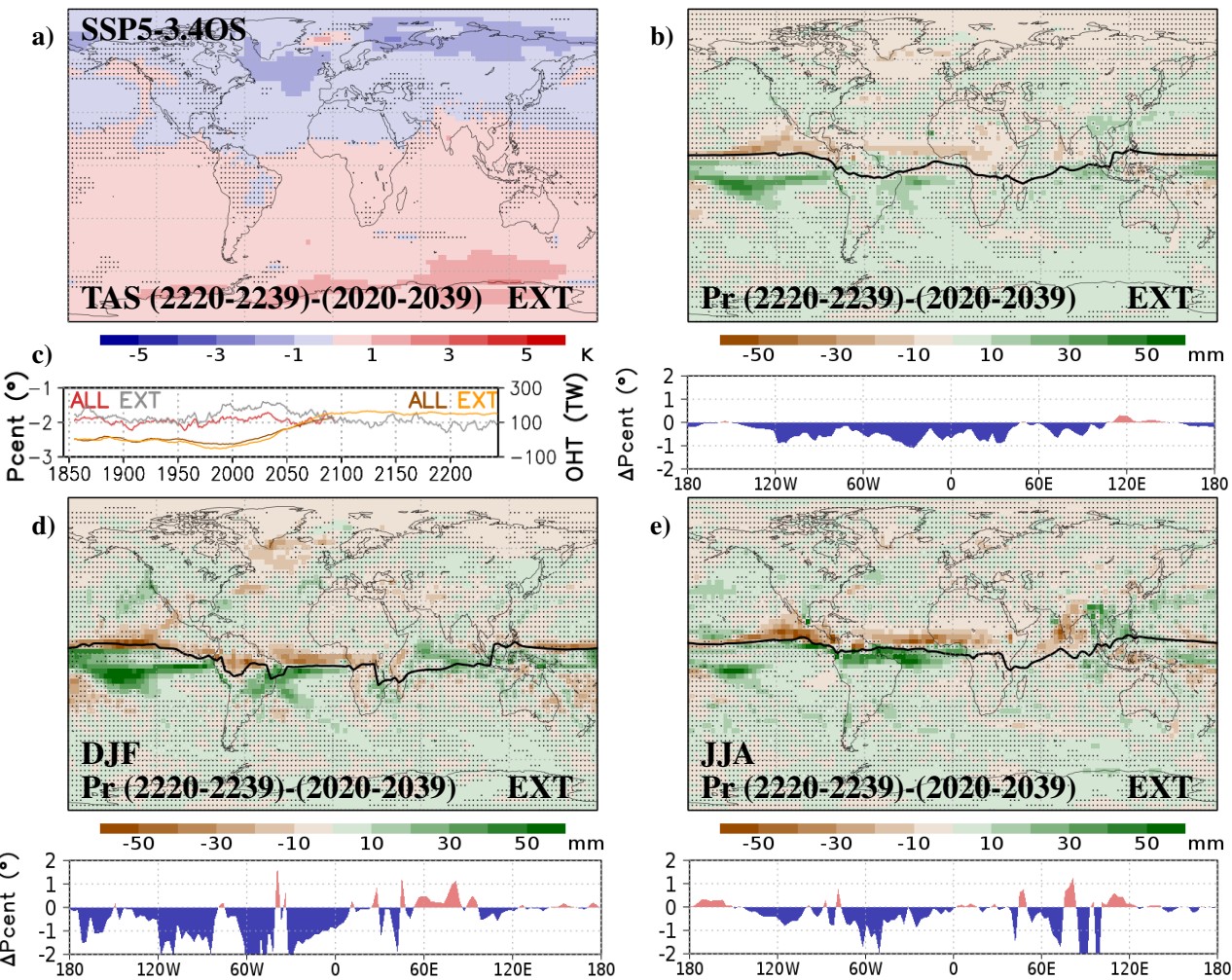

**Figure 5. (a)** Difference between the ensemble mean, temporal average values of surface air temperature (TAS) for the periods 2220-2239 and 2020-2039, obtained with the EXT ensemble of SSP5-3.4OS simulations. **(b)** Difference between the ensemble mean, temporal average values of annual precipitation (Pr) for the periods 2220-2239 and 2020-2039, obtained with the EXT ensemble of SSP5-3.4OS simulations. The ITCZ for the period 2020-2039, computed with the precipitation centroid, is included within the map, and difference between the precipitation centroid in 2220-2239 and 2020-2039, expressed in degrees of latitude, is included below. Stippling indicates locations where the differences are not significant (t-test with p<0.05). **(c)** Average position of the Atlantic ITCZ (70° W - 25° E), computed with the precipitation centroid (left axis), and average southwards Ocean Heat Transport (OHT) in the Atlantic basin (right axis), including both EXT and ALL ensembles. For the OHT, the same simulations as for temperature and precipitation have been considered, except for ACCESS-CM2, not providing this variable. **(d)** Same as (b), but for DJF. **(e)** Same as (b), but for JJA.

Fig. 8), which stabilize after the overshoot with warmest temperatures up to 1°C higher than those of 2034 (year before

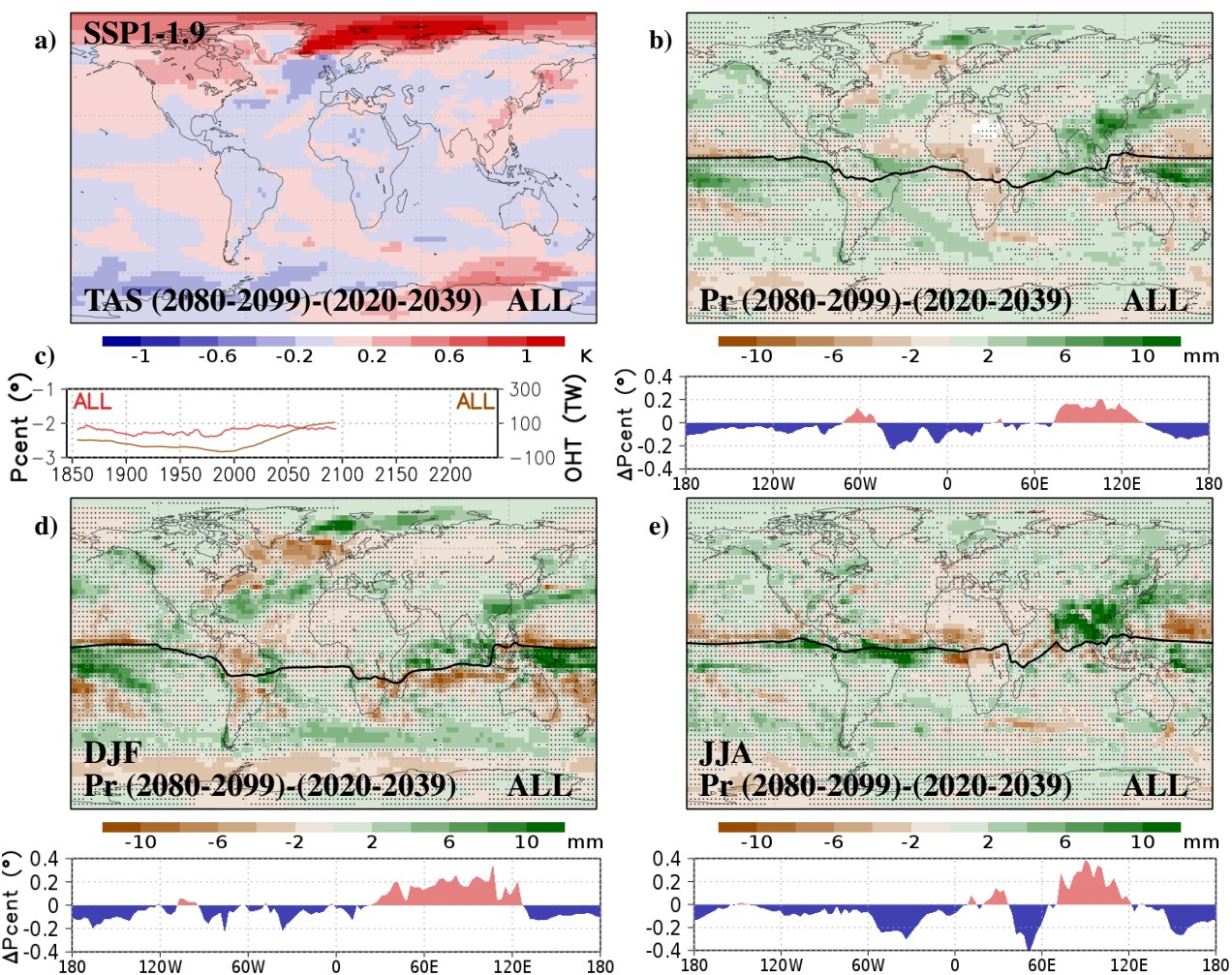

**Figure 6. (a)** Difference between the ensemble mean, temporal average values of surface air temperature (TAS) for the periods 2080-2099 and 2020-2039, obtained with the ALL ensemble of SSP1-1.9 simulations. **(b)** Difference between the ensemble mean, temporal average values of annual precipitation (Pr) for the periods 2080-2099 and 2020-2039, obtained with the ALL ensemble of SSP1-1.9 simulations. The ITCZ for the period 2020-2039, computed with the precipitation centroid, is included within the map, and difference between the precipitation centroid in 2080-2099 and 2020-2039, expressed in degrees of latitude, is included below. Stippling indicates locations where the differences are not significant (t-test with p<0.05). **(c)** Average position of the Atlantic ITCZ (70° W - 25° E), computed with the precipitation centroid (left axis), and average southwards Ocean Heat Transport (OHT) in the Atlantic basin (right axis). For the OHT, the same simulations as for temperature and precipitation have been considered, except for EC-Earth3, GFDL-ESM4, MIROC6, and MIROC-ES2L, not providing this variable. **(d)** Same as (b), but for DJF. **(e)** Same as (b), but for JJA.

the overshoot corresponding to the same global temperature as in 2300). On the contrary, an important decrease of colder temperatures is found in continental and high-latitude areas of Europe and Asia (NEU, EEU, WSB, ESB, and RAR; Fig. 8),

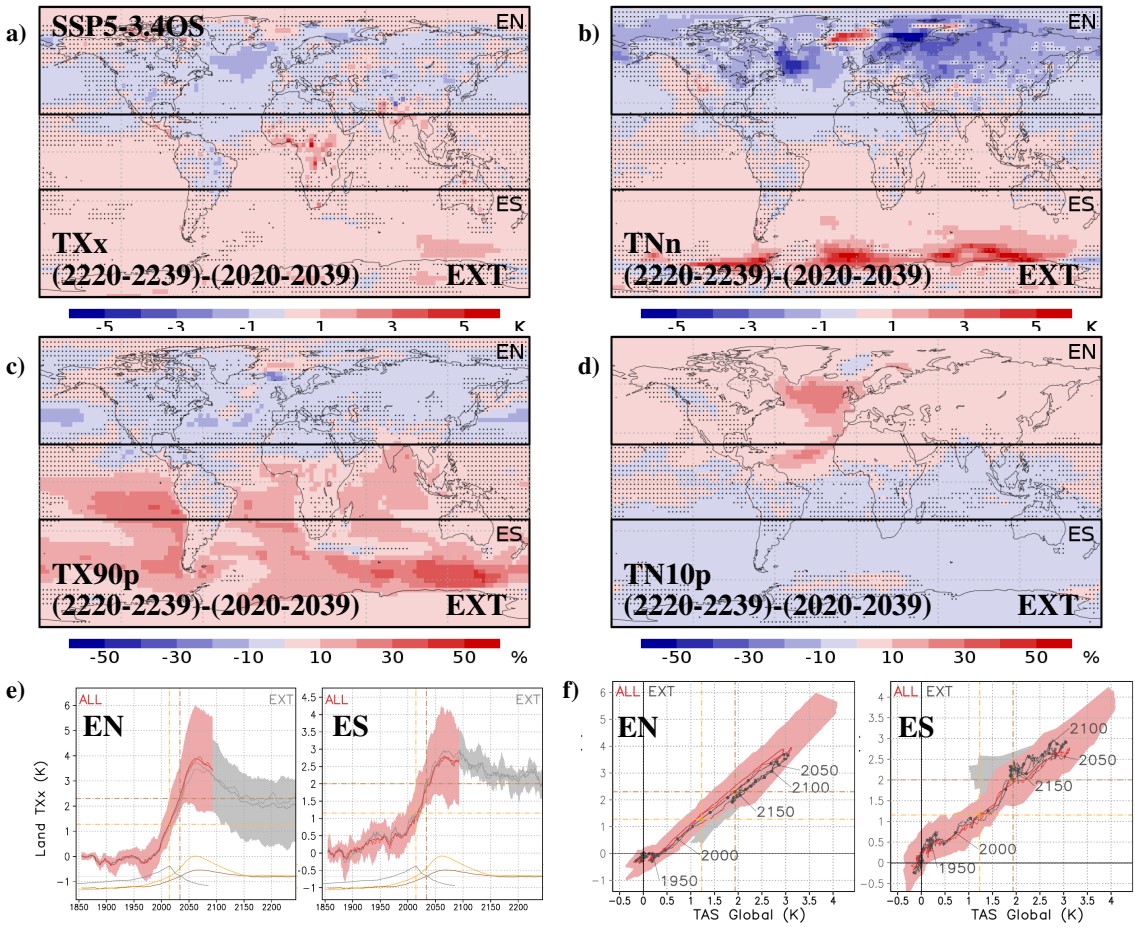

**Figure 7. (a-d)** Difference between the ensemble mean, temporal average values of extreme indices **(a)** TXx, **(b)** TNn, **(c)** TX90p, and **(d)** TN10p for the periods 2220-2239 and 2020-2039, obtained with the EXT ensemble of SSP5-3.4OS simulations. Stippling indicates locations where the differences are not significant (t-test with p<0.05). **(e)** Average (solid line) and median (dashed line) of TXx anomaly with respect to 1861-1880 obtained from the SSP5-3.4OS simulations for the extratropical areas of the NH (EN; 23° N - 90° N) and the extratropical areas of the SH (ES; 90° S - 23° S). Yellow, gray and brown curves in the lower part of the figure respectively show the $CO_2$ concentration from Meinshausen et al. (2020), the anthropogenic aerosol emissions (BC and OC) from Feng et al. (2020), and the global temperature obtained with the EXT ensemble. The vertical lines show the year before the overshoot with the same $CO_2$ concentration and global temperature as at the end of the run (2300 for the EXT ensemble of SSP5-3.4OS), while the horizontal lines represent the value of the average TXx in the EXT ensemble for those years. **(f)** Average (solid line) and median (dashed line) of TXx anomaly with respect to 1861-1880 obtained from the SSP5-3.4OS simulations for EN and ES with respect to the global average of surface air temperature (TAS). Yellow and brown lines show the values of TXx and global TAS in the years before the overshoot with the same $CO_2$ concentration and global temperature as at the end of the run (2300 for the EXT ensemble of SSP5-3.4OS).

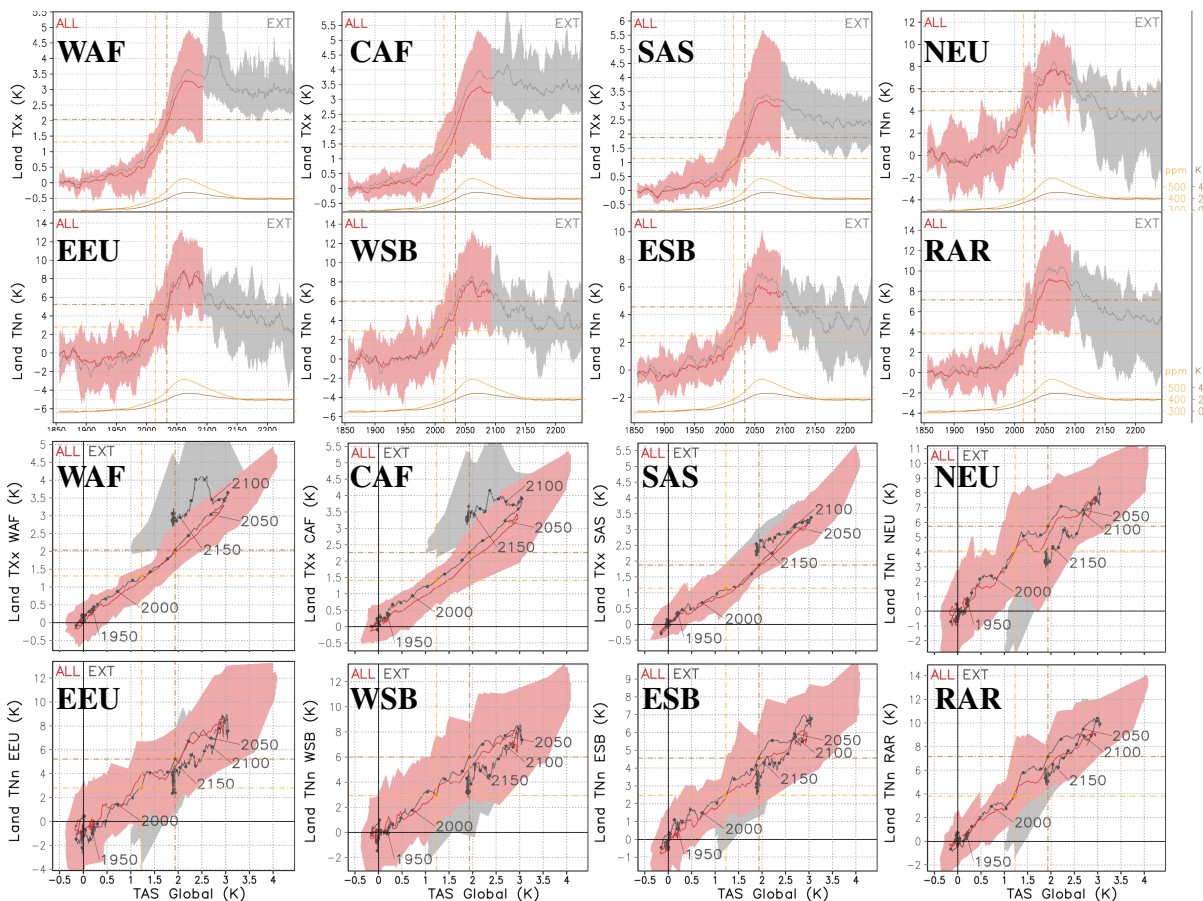

**Figure 8.** Regional average of TNn and TXx anomaly with respect to 1861-1880, over time (top 2 rows) and with respect to the global average of surface air temperature (bottom 2 rows), obtained from the SSP5-3.4OS simulations for a set of IPCC reference regions (Fig. 1), including TXx for WAF, CAF, and SAS; and TNn for NEU, EEU, WSB, ESB, and RAR. Yellow and brown curves in the lower part of each figure respectively show the $CO_2$ concentration from Meinshausen et al. (2020) and the global temperature obtained with the EXT ensemble. The vertical lines show the year before the overshoot with the same $CO_2$ concentration and global temperature as at the end of the run (2300 for the EXT ensemble of SSP5-3.4OS), while the horizontal lines represent the value of the average index in the EXT ensemble for those years.

showing coldest temperatures up to 3°C lower than those of 2034, associated with a more frequent occurrence of cold extremes like TN10p (Fig. 7d). Even if changes in average temperatures are generally limited to 2°C, changes in TXx reach 3°C and, for some high-latitude regions, changes in TNn reach 5°C with respect to the pre-overshoot situation. This opposite behavior between extratropical areas of NH and SH is evident in Fig. 7e, where the average of the EXT ensemble stabilizes to a warmest temperature lower than that of 2034 (year before the overshoot corresponding to the same global temperature as in 2300) for the extratropical areas of the NH (EN; Fig. 7e), and higher for the extratropical areas of the SH (ES; Fig. 7e).

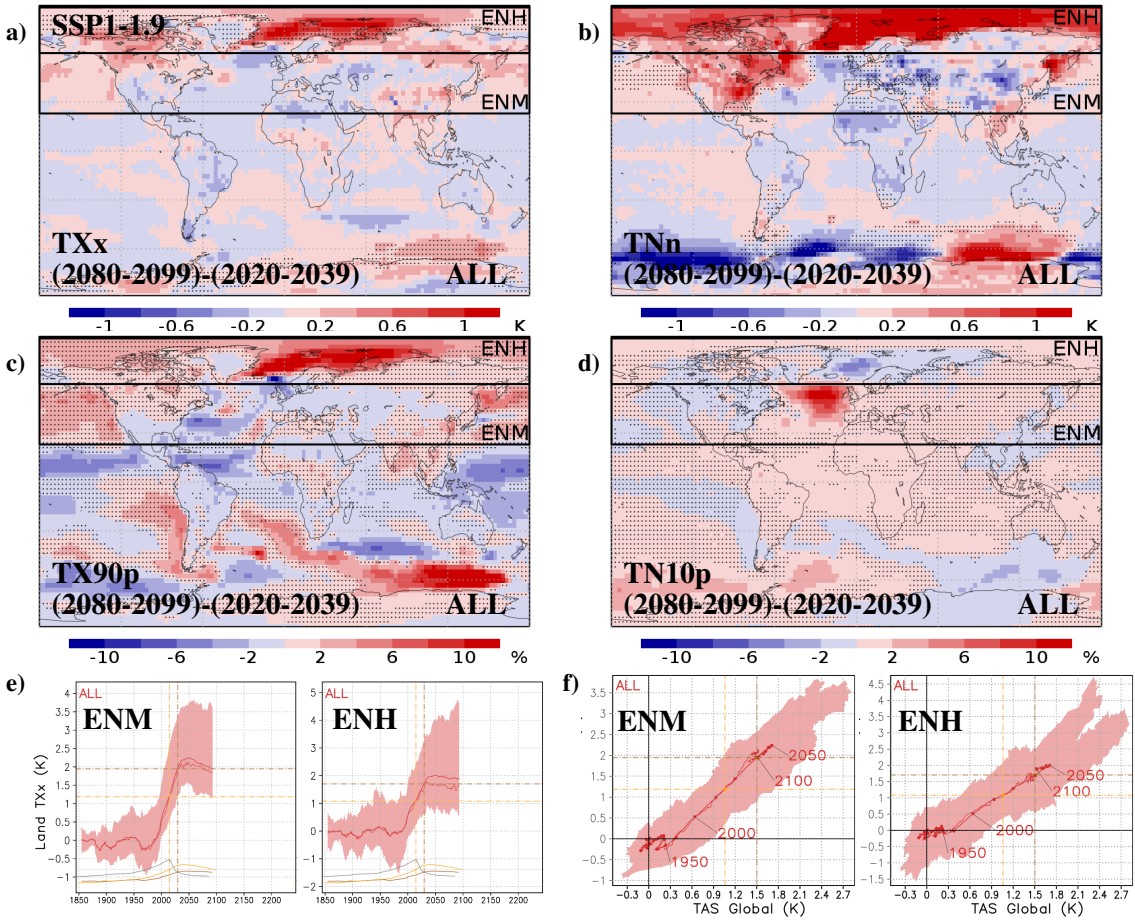

**Figure 9. (a-d)** Difference between the ensemble mean, temporal average values of extreme indices **(a)** TXx, **(b)** TNn, **(c)** TX90p, and **(d)** TN10p for the periods 2080-2099 and 2020-2039, obtained with the ALL ensemble of SSP1-1.9 simulations. Stippling indicates locations where the differences are not significant (t-test with p<0.05). **(e)** Average (solid line) and median (dashed line) of TXx anomaly with respect to 1861-1880 obtained from the SSP1-1.9 simulations for the mid-latitude extratropical areas of the NH (ENM; 23° N - 60° N) and the high-latitude extratropical areas of the NH (ENH; 60° N - 90° N). Yellow, gray and brown curves in the lower part of the figure respectively show the CO$_2$ concentration from Meinshausen et al. (2020), the anthropogenic aerosol emissions (BC and OC) from Feng et al. (2020), and the global temperature obtained with the ALL ensemble. The vertical lines show the year before the overshoot with the same CO$_2$ concentration and global temperature as at the end of the run (2100 for the ALL ensemble of SSP1-1.9), while the horizontal lines represent the value of the average TXx in the ALL ensemble for those years. **(f)** Average (solid line) and median (dashed line) of TXx anomaly with respect to 1861-1880 obtained from the SSP1-1.9 simulations for ENM and ENH with respect to the global average of surface air temperature (TAS). Yellow and brown lines show the values of TXx and global TAS in the years before the overshoot with the same CO$_2$ concentration and global temperature as at the end of the run (2100 for the ALL ensemble of SSP1-1.9).

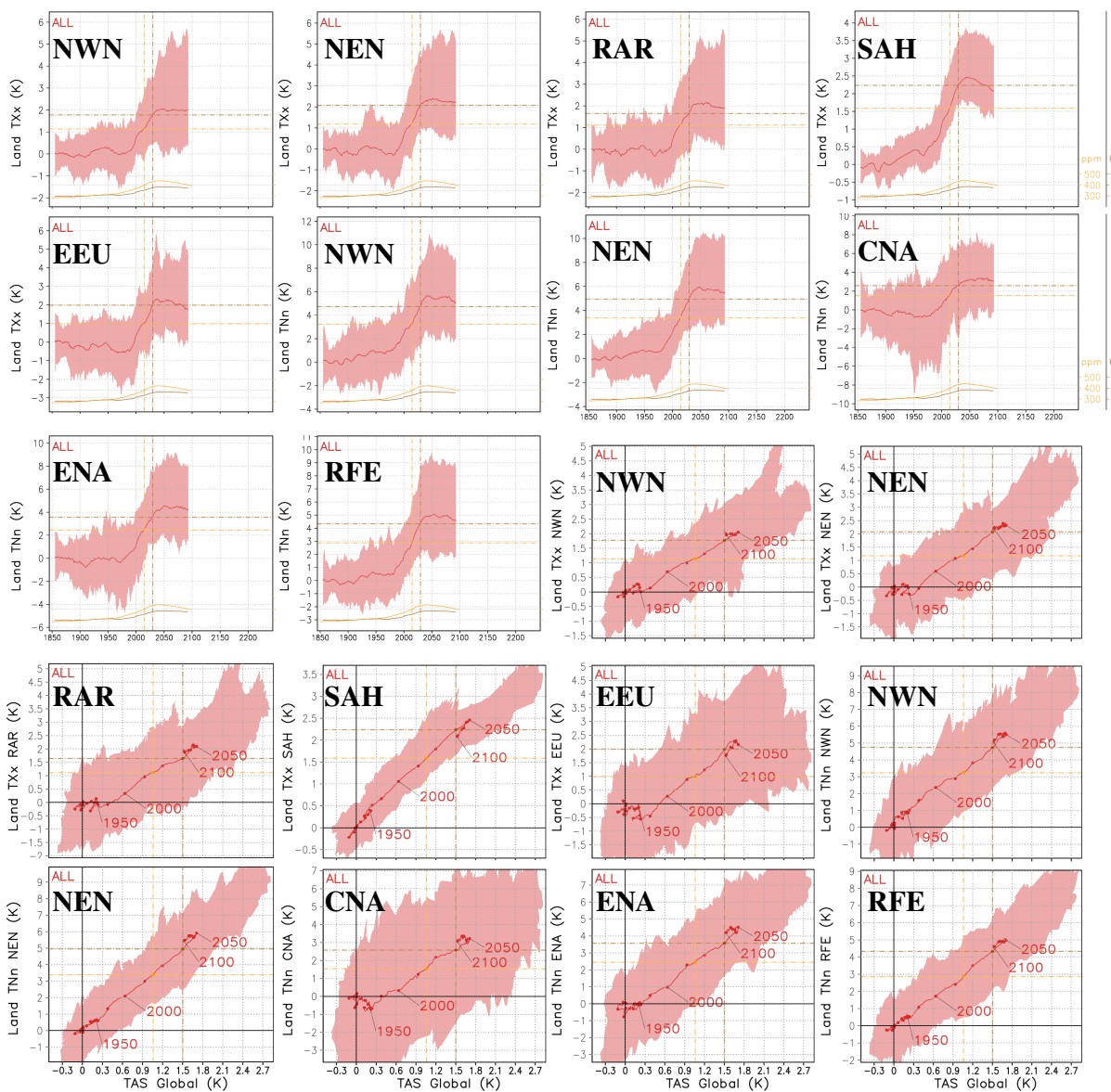

**Figure 10.** Regional average of TNn and TXx anomaly with respect to 1861-1880, over time (top 3 rows) and with respect to the global average of surface air temperature (bottom 3 rows), obtained from the SSP1-1.9 simulations for a set of IPCC reference regions (Fig. 1), including TXx for NWN, NEN, RAR, SAH, and EEU; and TNn for NWN, NEN, CNA, ENA, and RFE. Yellow and brown curves in the lower part of each figure respectively show the $CO_2$ concentration from Meinshausen et al. (2020) and the global temperature obtained with the ALL ensemble. The vertical lines show the year before the overshoot with the same $CO_2$ concentration and global temperature as at the end of the run (2100 for the ALL ensemble of SSP1-1.9), while the horizontal lines represent the value of the average index in the ALL ensemble for those years.

Figure 7f show that the relationship between global temperature and regional TXx remains the same before and after the overshoot, with persistent changes mostly produced around the maximum. From 2000 to 2060, TXx increases linearly with respect to the global average of temperature, both for EN and EH. From 2060 to 2080, there is a transition period in which TXx is decoupled from the global temperature. During this period, TXx decreases for EN and increases for ES (Fig. 7f), potentially linked to the timing of the regional maximum of temperature, reached before the global maximum for most continental areas of the NH and after the global maximum for large areas of the SH (Fig. 4a). The scaling of regional extremes with the global mean temperature is recovered afterwards, and starting from 2100 TXx decreases linearly with respect to the global average of temperature, both for EN and EH (Fig. 7f), with a slope similar to that of the increasing phase between 2000 and 2060, but vertically shifted keeping the TXx differences cumulated during the transition period.

For the case of SSP1-1.9, changes in temperature extremes with respect to the pre-overshoot situation only reach 1°C for certain high-latitude regions (Fig. 9a), consistent with the higher average temperatures found after overshoot for these areas (Fig. 6a). The higher TXx compared to the pre-overshoot situation is particularly relevant in northern North America and northern Asia (NWN, NEN, and RAR; Fig. 10), while TNn increases the most in areas of northern and central North America and northeastern Asia (NWN, NEN, CNA, ENA, and RFE; Fig. 10). Conversely, a decrease of TXx and TNn compared to the pre-overshoot situation is found in mid-latitude regions like the Sahara and eastern Europe (SAH and EEU; Fig. 10). This opposition between high and mid latitudes is illustrated in Fig. 9e, where the TXx for the ALL ensemble reaches in 2100 values that are below those of 2030 (year before the overshoot corresponding to the same global temperature as in 2100) for mid latitudes (ENM; Fig. 9e) and above for high latitudes (ENH; Fig. 9e). These results are however limited by the fact that simulations of SSP1-1.9 only extend up to 2100, not reaching stabilization.

Both for SSP5-3.4OS (Fig. 7) and SSP1-1.9 (Fig. 9), similar results are obtained when using either of the ensemble average and ensemble median, confirming that the average is not biased by any individual model. A more detailed analysis on the differences between models is included in Appendix A.

## 3.4 Changes in extreme precipitation

Consistent with changes in mean precipitation, precipitation extremes for SSP5-3.4OS (Fig. 11a-b) are impacted by the ITCZ shifts found in Fig. 5b. Areas north of the ITCZ, like Western and Central Africa (WAF and CAF; Fig. 12) show a decline of precipitation extremes compared to the pre-overshoot situation, while areas south of the ITCZ, like Madagascar and the southern part of Indonesia (MDG and SEA; Fig. 12) show an increase. This opposite behavior between areas north and south of the ITCZ is summarized in Fig. 11c, where the average of the EXT ensemble stabilizes to Rx5day values lower than those of 2034 (year before the overshoot corresponding to the same global temperature as in 2300) for the areas north of the ITCZ (TN; Fig. 11c), and higher for the areas to the south (TS; Fig. 11c). As for the case of TXx, the relationship between global temperature and regional Rx5day remains the same before and after the overshoot but the trajectory is shifted up (TS) or down (TN) compared to the pre-overshoot period (Fig. 11d), with the persistent changes mostly cumulated during the transition phase (from 2060 to 2080).

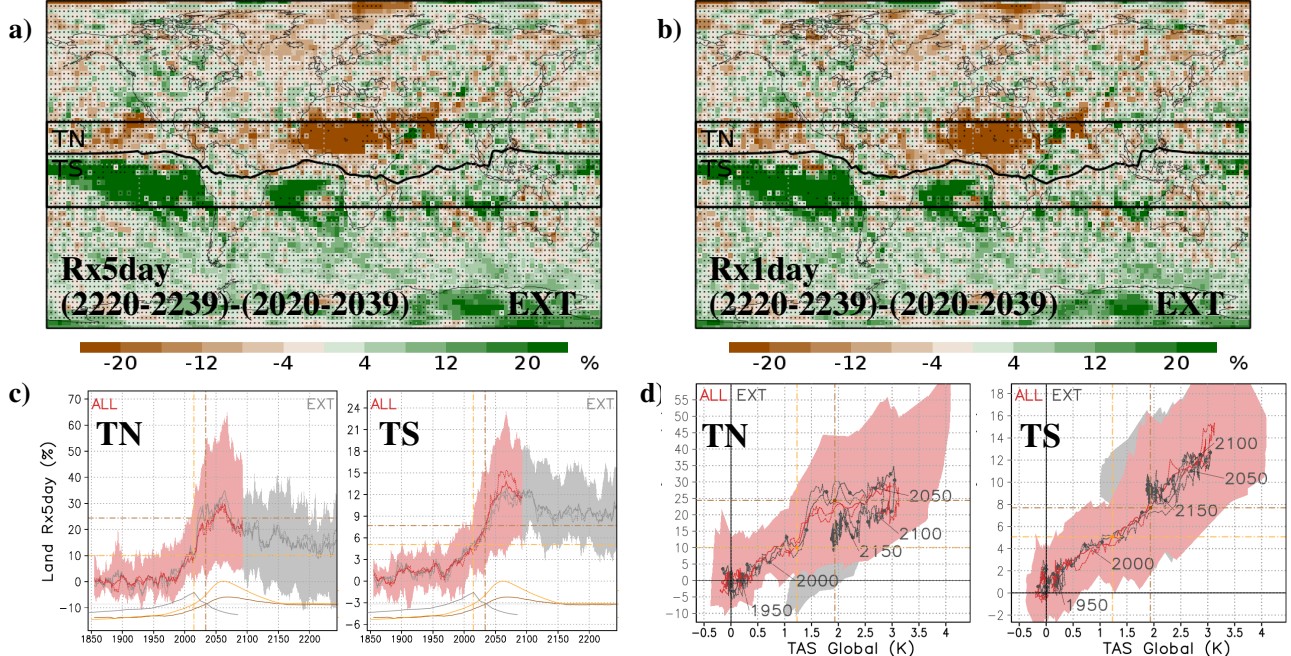

**Figure 11. (a-b)** Difference between the ensemble mean, temporal average values of extreme indices **(a)** Rx5day and **(b)** Rx1day for the periods 2220-2239 and 2020-2039, obtained with the EXT ensemble of SSP5-3.4OS simulations. Rx5day and Rx1day are expressed in percentage of variation with respect to 1861-1880. Stippling indicates locations where the differences are not significant (t-test with $p<0.05$). **(e)** Average (solid line) and median (dashed line) of Rx5day percentage of variation with respect to 1861-1880 obtained from the SSP5-3.4OS simulations for the tropical areas north of the 2020-2039 ITCZ (TN; ITCZ - $23°$ N) and the tropical areas south of the 2020-2039 ITCZ (TS; $23°$ S - ITCZ). Yellow, gray and brown curves in the lower part of the figure respectively show the $CO_2$ concentration from Meinshausen et al. (2020), the anthropogenic aerosol emissions (BC and OC) from Feng et al. (2020), and the global temperature obtained with the EXT ensemble. The vertical lines show the year before the overshoot with the same $CO_2$ concentration and global temperature as at the end of the run (2300 for the EXT ensemble of SSP5-3.4OS), while the horizontal lines represent the value of the average Rx5day in the EXT ensemble for those years. **(f)** Average (solid line) and median (dashed line) of Rx5day percentage of variation with respect to 1861-1880 obtained from the SSP5-3.4OS simulations for TN and TS with respect to the global average of surface air temperature (TAS). Yellow and brown lines show the values of Rx5day and global TAS in the years before the overshoot with the same $CO_2$ concentration and global temperature as at the end of the run (2300 for the EXT ensemble of SSP5-3.4OS).

For SSP1-1.9, Fig. 13a-b show that ITCZ shifts, even if smaller than those of SSP5-3.4OS, also explain changes in Rx5day and Rx1day. Areas north of the Atlantic ITCZ like Western and Central Africa (WAF and CAF; Fig. 14) show a decline of precipitation extremes with respect to the pre-overshoot climate. The fact that the Atlantic and Pacific ITCZ shifts to the south while Indian ITCZ shifts to the north (Fig. 6) makes the behavior not so clear for Southeast Asia. On regions around the Atlantic and Pacific ITCZ, the ALL ensemble reaches in 2100 values that are below those of 2030 (year before the overshoot corresponding to the same global temperature as in 2100) for areas to the north (TNW; Fig. 13c) and similar values to those

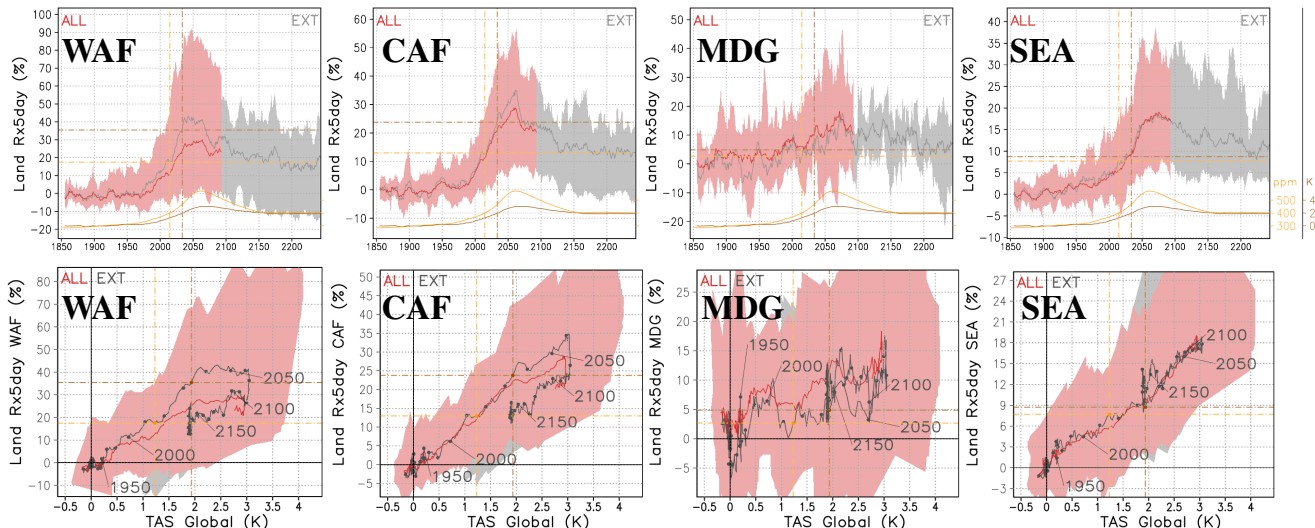

**Figure 12.** Regional average of Rx5day percentage of variation with respect to 1861-1880, over time (top row) and with respect to the global average of surface air temperature (bottom row), obtained from the SSP5-3.4OS simulations for a set of IPCC reference regions (Fig. 1), including WAF, CAF, MDG, and SEA. Yellow and brown curves in the lower part of each figure respectively show the $CO_2$ concentration from Meinshausen et al. (2020) and the global temperature obtained with the EXT ensemble. The vertical lines show the year before the overshoot with the same $CO_2$ concentration and global temperature as at the end of the run (2300 for the EXT ensemble of SSP5-3.4OS), while the horizontal lines represent the value of the average index in the EXT ensemble for those years.

of 2030 for areas to the south (TNS; Fig. 13c), consistent with a southward shift of the ITCZ. As for the case of temperature extremes, these results are limited by the fact that for most regions precipitation is not fully stabilized by 2100.

## 4   Discussion and Conclusions

Our analysis of CMIP6 overshoot scenarios show a relevant impact of large-scale mechanisms on generating non-symmetric changes at regional scales during global temperature increases and decreases. This impact is particularly strong under strong forcing conditions like those of SSP5-3.4OS, but is also relevant in weaker overshoots like that of SSP1-1.9. For both scenarios, the situation after the overshoot differs from that of before in a significant way, both in terms of temperature and precipitation spatial distributions.

For SSP5-3.4OS, the situation after the overshoot is characterized by a colder NH and a warmer SH, associated with a southward shift of the ITCZ, in line with the results in idealized experiments (Kug et al., 2022). The analysis of SSP1-1.9 is limited by the period covered by simulations. Even if the maximum of temperature for this experiment is reached for most regions before 2050, the climate is not fully stabilized by 2100, when the simulations end. Even with that, the analysis of the final state shows significant differences with respect to the situation before the overshoot, with higher temperatures for polar

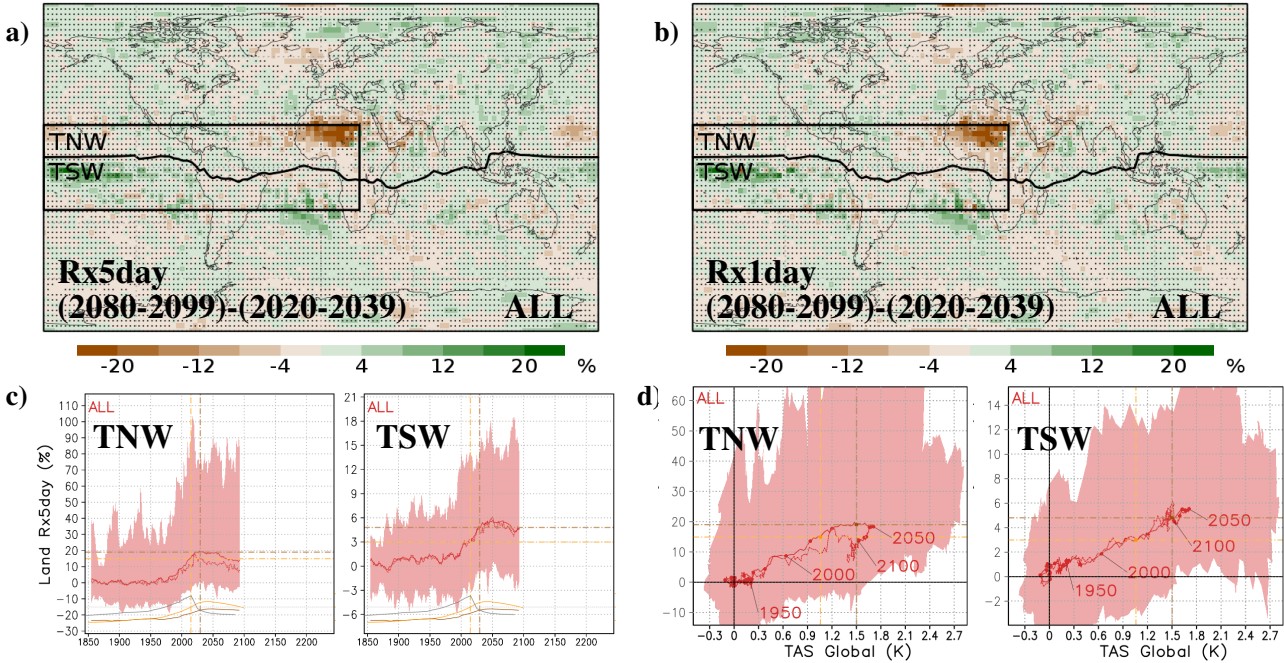

**Figure 13. (a-b)** Difference between the ensemble mean, temporal average values of extreme indices **(a)** Rx5day and **(b)** Rx1day for the periods 2080-2099 and 2020-2039, obtained with the ALL ensemble of SSP1-1.9 simulations. Rx5day and Rx1day are expressed in percentage of variation with respect to 1861-1880. Stippling indicates locations where the differences are not significant (t-test with p<0.05). **(c)** Average (solid line) and median (dashed line) of Rx5day percentage of variation with respect to 1861-1880 obtained from the SSP1-1.9 simulations for the tropical areas north of the 2020-2039 Atlantic and eastern Pacific ITCZ (TNW; ITCZ - 23° N; 180° W - 25° E) and the tropical areas south of the 2020-2039 Atlantic and eastern Pacific ITCZ (TSW; 23° S - ITCZ; 180° W - 25° E). Yellow, gray and brown curves in the lower part of the figure respectively show the $CO_2$ concentration from Meinshausen et al. (2020), the anthropogenic aerosol emissions (BC and OC) from Feng et al. (2020), and the global temperature obtained with the ALL ensemble. The vertical lines show the year before the overshoot with the same $CO_2$ concentration and global temperature as at the end of the run (2100 for the ALL ensemble of SSP1-1.9), while the horizontal lines represent the value of the average Rx5day in the ALL ensemble for those years. **(f)** Average (solid line) and median (dashed line) of Rx5day percentage of variation with respect to 1861-1880 obtained from the SSP1-1.9 simulations for TNW and TSW with respect to the global average of surface air temperature (TAS). Yellow and brown lines show the values of Rx5day and global TAS in the years before the overshoot with the same $CO_2$ concentration and global temperature as at the end of the run (2100 for the ALL ensemble of SSP1-1.9).

regions of the NH and for certain areas of the Southern Ocean, and with ITCZ shifts, to the south over the Pacific and Atlantic basin and to the north over the Indian basin.

    Changes in temperature and precipitation during the overshoot may explain relevant changes and hysteresis in regional extremes. Warmest regional temperatures after overshoot exceed those obtained at the same global average temperature before the overshoot for most tropical and extratropical regions of the SH in SSP5-3.4OS and for high-latitude regions both of the

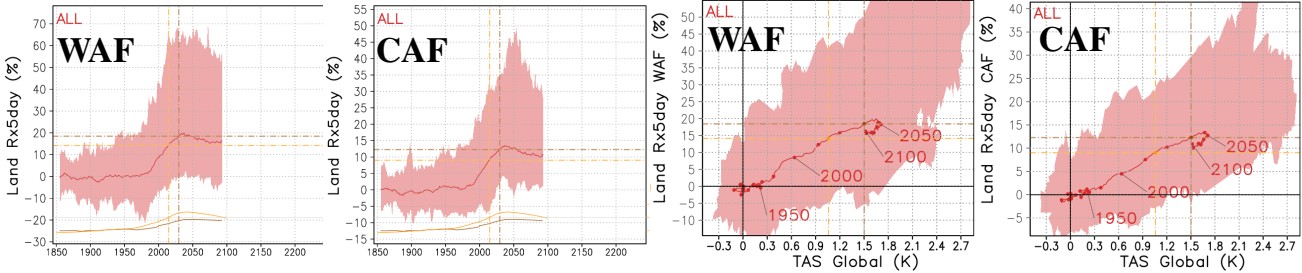

**Figure 14.** Regional average of Rx5day percentage of variation with respect to 1861-1880, over time (right) and with respect to the global average of surface air temperature (left), obtained from the SSP1-1.9 simulations for the IPCC reference regions WAF and CAF (Fig. 1). Yellow and brown curves in the lower part of each figure respectively show the $CO_2$ concentration from Meinshausen et al. (2020) and the global temperature obtained with the EXT ensemble. The vertical lines show the year before the overshoot with the same $CO_2$ concentration and global temperature as at the end of the run (2100 for the ALL ensemble of SSP1-1.9), while the horizontal lines represent the value of the average index in the ALL ensemble for those years.

NH and SH in SSP1-1.9. This is consistent with, and can explain, the partially reversed behavior found by Pfleiderer et al. (2024) in 2100 for the TXx of RAR, NEU, GIC, NEN, NZ, and SSA (for the region definitions see Fig. 1). The persistent changes are even larger for the coldest temperatures, showing a significant decline in many continental regions of the NH both for SSP5-3.4OS and SSP1-1.9. This was also found by Pfleiderer et al. (2024) for the TNn of WCA, SAH, and TIB, with a partially reversed behavior in 2100, but not so clearly for other regions like MED, WCE, and EEU, where the stabilization is

reached after 2100 (Fig. 4d). Despite the minor role of hysteresis found by Walton and Huntingford (2024) for the regional precipitation of tropical areas, a relevant role is found in regions around the ITCZ. Precipitation extremes for these regions are impacted by ITCZ shifts, with both experiments showing a decline in the intensity of extreme precipitation in regions to the north of the ITCZ, like Western and Central Africa, in line with the overcompensated behavior found by Pfleiderer et al. (2024) for these regions.

For SSP5-3.4OS, the fact that the maximum of regional temperatures is reached before 2070 for most continental areas and after 2090 for the Southern Ocean suggests a relevant role of the inertia of the ocean, experiencing warming and cooling phases delayed compared to those of continental areas. However, other mechanisms like changes in the AMOC (Moreno-Chamarro et al., 2020) or changes in sea ice (Li et al., 2020) may also contribute. For SSP1-1.9, showing less asymmetry between NH and SH and a more intense contrast between high and mid latitudes, a larger role of anthropogenic aerosol emissions and ice

melting may be present, generating persistent changes in polar regions during the overshoot.

Changes in the relationship between the regional climate conditions and the global mean temperature mainly take place during the transition period around the global temperature maximum (from 2060 to 2080 for SSP5-3.4OS and from 2040 to 2060 for SSP1-1.9). Afterwards, the relationship between global mean temperatures and regional extremes recovers a similar slope to that of the pre-overshoot period, but with an offset cumulated during the transition phase. The evolution of regional

extremes is mostly coupled to the evolution of the global temperatures during the periods of increasing and decreasing global

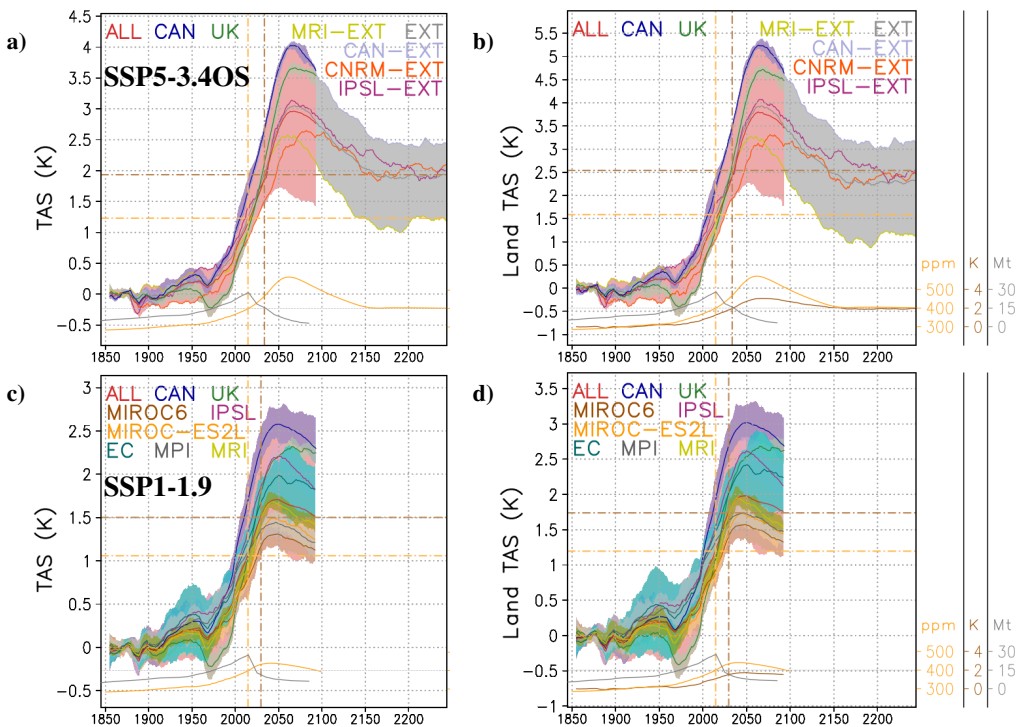

**Figure A1.** Same as Fig. 2, but including the average of the ensembles from CanESM5 (CAN), UKESM1-0-LL (UK), MIROC6, MIROC-ES2L, IPSL-CM6A-LR (IPSL), EC-Earth3 (EC), MPI-ESM1-2-LR (MPI), and MRI-ESM2-0 (MRI), and, for SSP5-3.4OS, the extended simulations of CanESM5 (CAN-EXT), CNRM-ESM2-1 (CNRM-EXT), IPSL-CM6A-LR (IPSL-EXT), and MRI-ESM2-0 (MRI-EXT).

temperature, but it is decoupled during the transition period around the global maximum depending on the timing of regional maximum temperatures, generating region-dependent irreversibilities.

These persistent changes may be linked to a different timing of the regional temperature maximum. Areas with an anticipated maximum, like most continental areas of the NH, tend to show mitigated warm extremes and more intense cold extremes, and conversely for areas with a delayed maximum, like the Southern Ocean and some land areas of the SH. The results of this work allow for a better understanding of the irreversibility of regional extremes, by linking it to large-scale mechanisms like temperature asymmetries and ITCZ shifts. They also allow identifying those regions more impacted by irreversibility processes, including those around the ITCZ, with particular impacts on precipitation extremes, and those in extratropical areas like North America, Europe and central Asia, with particular impacts on temperature extremes.

## Appendix A: Differences across models

Figure 2, 7, and 9 show results based on the average and the median of ALL and EXT ensembles. The dispersion of individual simulations is also included as a shading, but to analyze if this dispersion is due to differences across models or to internal

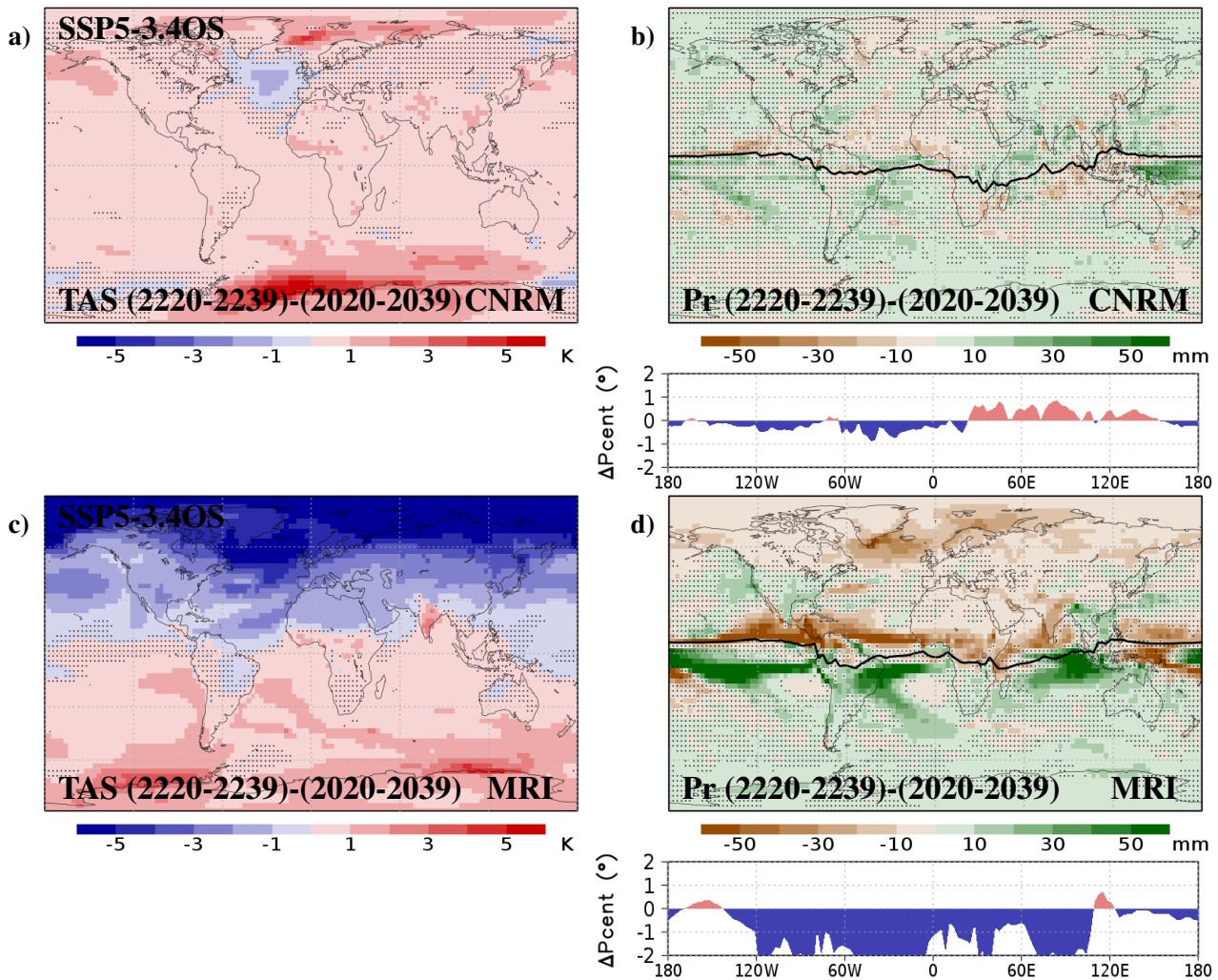

**Figure A2.** Same as Fig. 5a-b, but for the extended simulations of **(a-b)** CNRM-ESM2-1 (CNRM) and **(c-d)** MRI-ESM2-0 (MRI).

variability, the results are also presented for the ensemble mean of each individual model in Fig. A1, A3, and A5. For this, all the models providing several simulations has been considered, including CanESM5 and UKESM1-0-LL for SSP5-3.4OS and
CanESM5, EC-Earth3, IPSL-CM6A-LR, MIROC6, MIROC-ES2L, MPI-ESM1-2-LR, MRI-ESM2-0, and UKESM1-0-LL for SSP1-1.9. For SSP5-3.4OS, the individual simulations covering the period up to 2300 have been also included.

Figure A1 shows that each model simulates a different level of temperature change during the overshoot. Both for SSP5-3.4OS and for SSP1-1.9, CanESM5 and UKESM1-0-LL tend to show a temperature response larger than that of the ALL ensemble, while other models like MPI-ESM1-2-LR and MIROC6 show a more mitigated response. Despite these differences

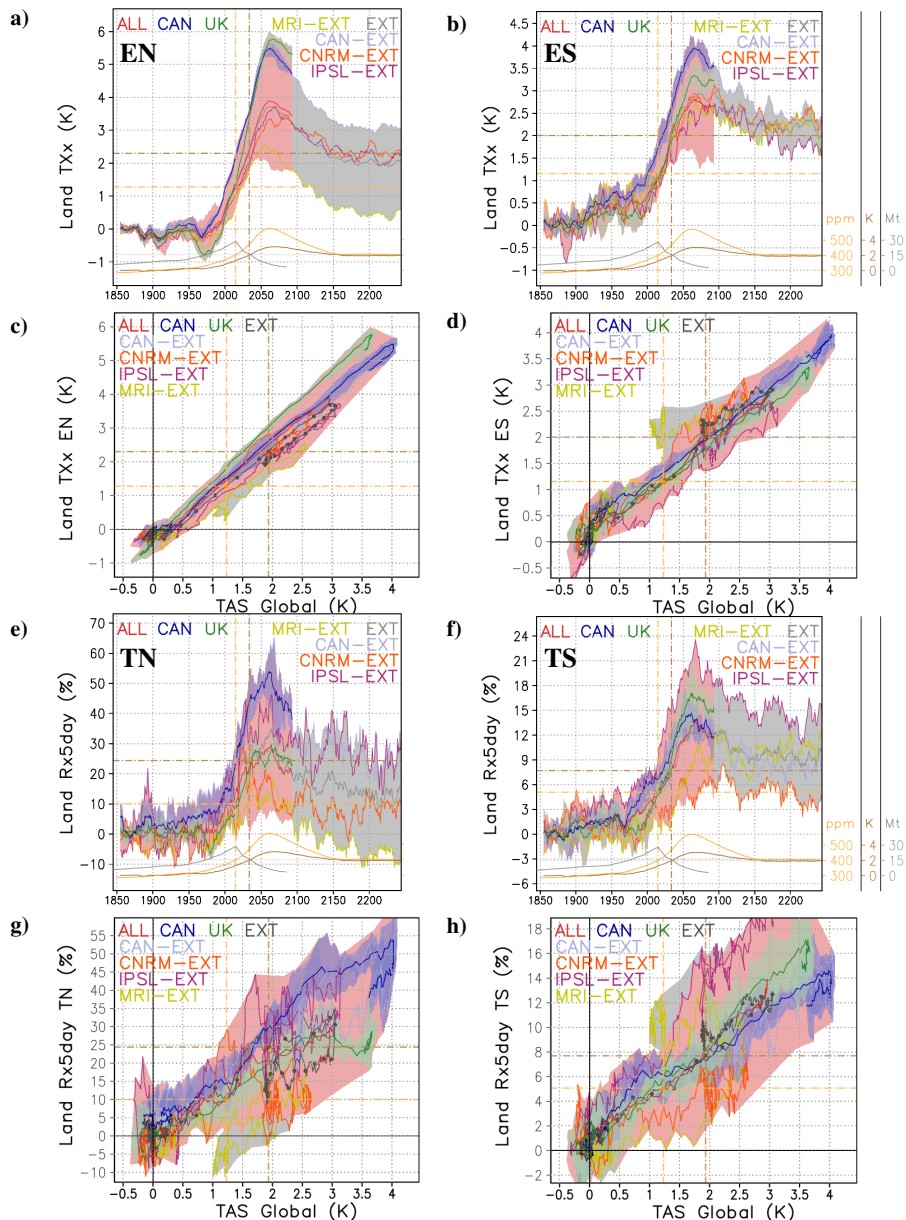

**Figure A3. (a-d)** Same as Fig. 7e-f, but including the average of the ensembles from CanESM5 (CAN) and UKESM1-0-LL (UK), and the extended simulations of CanESM5 (CAN-EXT), CNRM-ESM2-1 (CNRM-EXT), IPSL-CM6A-LR (IPSL-EXT), and MRI-ESM2-0 (MRI-EXT). **(e-h)** Same as Fig. 11c-d, but including the average of the ensembles from CanESM5 (CAN) and UKESM1-0-LL (UK), and the extended simulations of CanESM5 (CAN-EXT), CNRM-ESM2-1 (CNRM-EXT), IPSL-CM6A-LR (IPSL-EXT), and MRI-ESM2-0 (MRI-EXT).

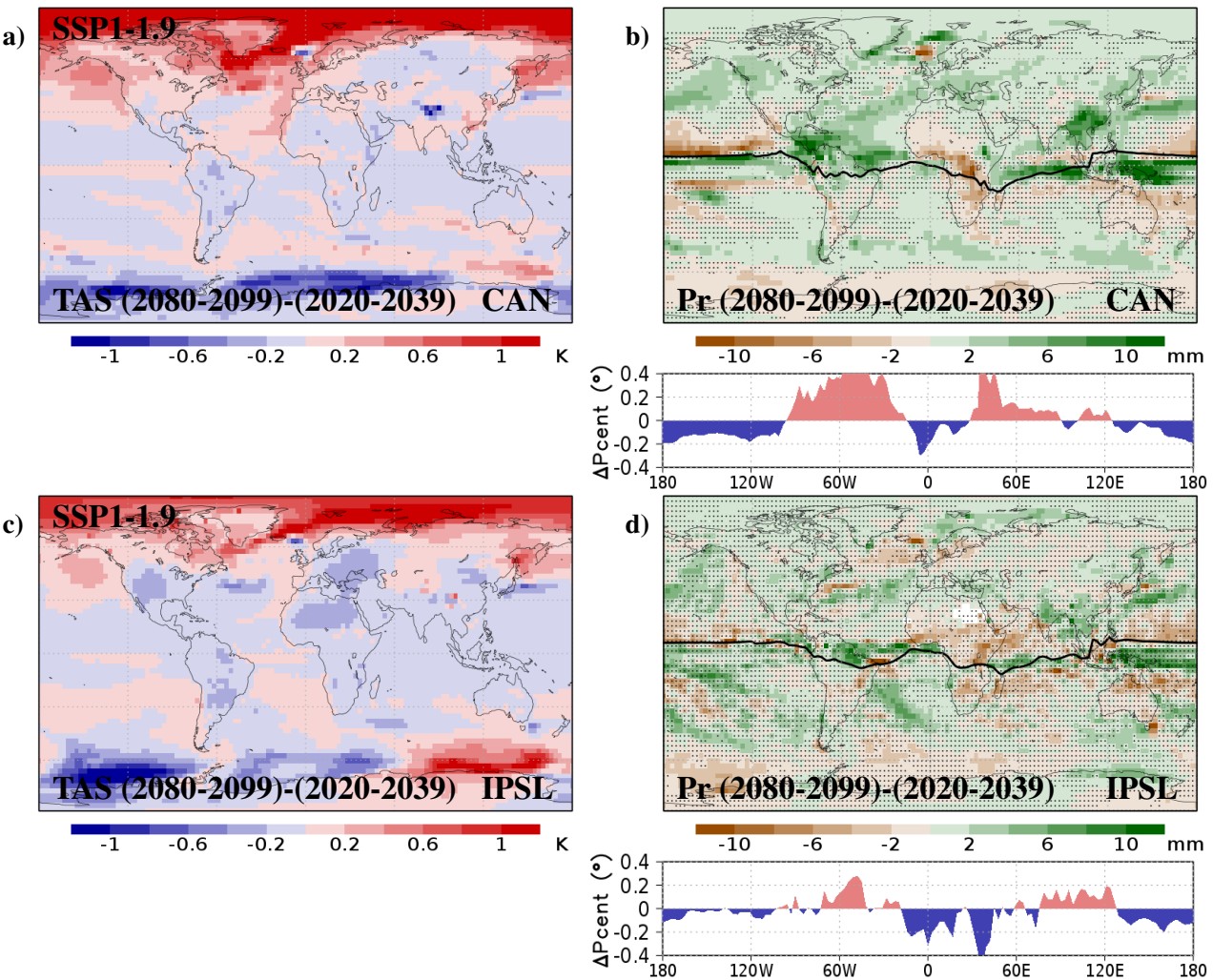

**Figure A4.** Same as Fig. 6a-b, but for the average of the ensembles from **(a-b)** CanESM5 (CAN) and **(c-d)** IPSL-CM6A-LR (IPSL).

in the level of temperature response, all the models show a similar temporal evolution of the global average of temperature, confirming their suitability for being combined in a single ALL ensemble.

Regarding the spatial patterns, Fig. A2 and A4 show the same differences as in Fig. 5 and 6 but considering only some particular model simulations. In particular, the extended simulations of CNRM-ESM2-1 and MRI-ESM2-0 are presented for SSP5-3.4OS (Fig. A2) and the ensemble averages of CanESM5 and IPSL-CM6A-LR are presented for SSP1-1.9 (Fig. A4). Despite the general agreement among the models on simulating a post-overshoot climate characterised by temperature asymmetries and ITCZ shifts, for the case of SSP5-3.4OS the spatial patterns and the magnitude of the changes strongly differ across the models (Fig. A2). Some models like MRI-ESM2-0 show a strong hemispherical temperature asymmetry between

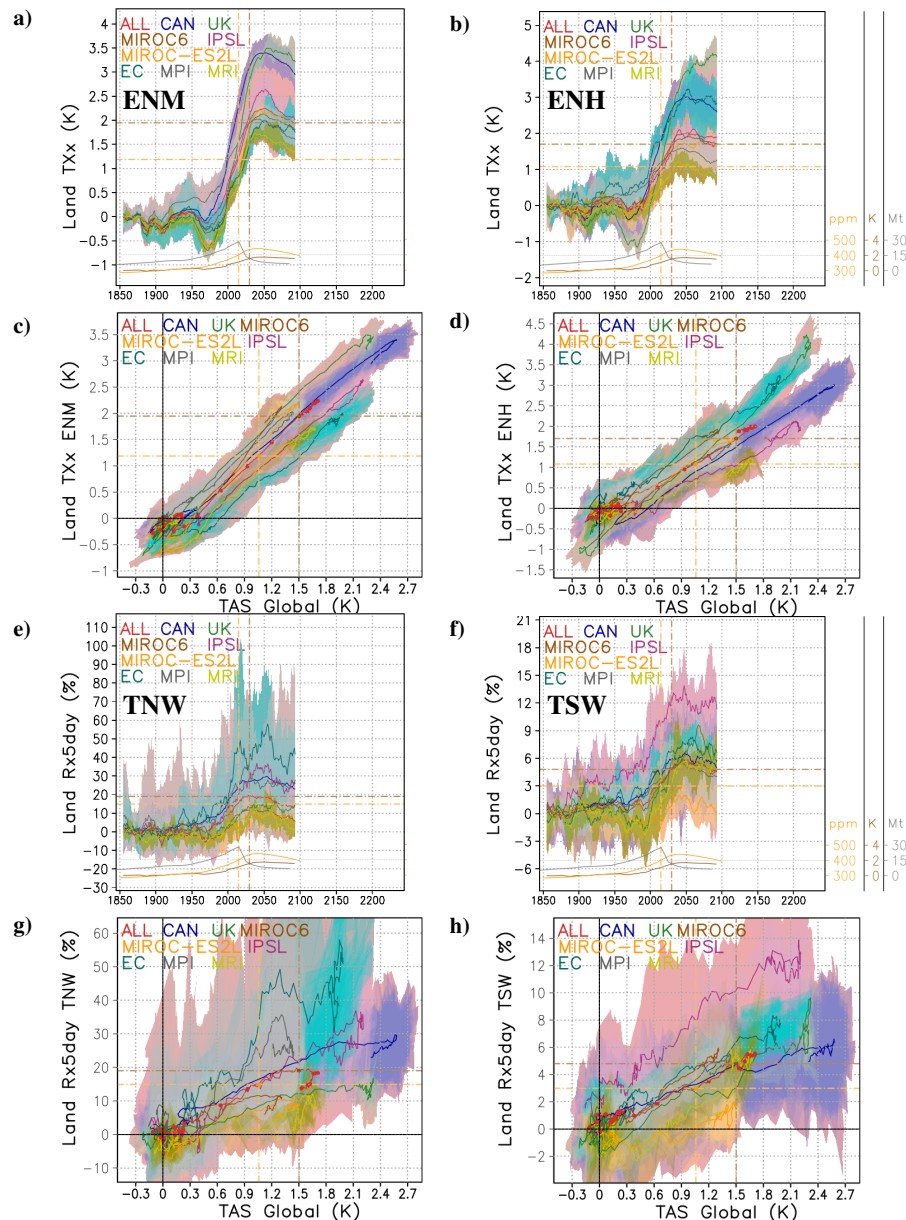

**Figure A5. (a-d)** Same as Fig. 9e-f, but including the average of the ensembles from CanESM5 (CAN), UKESM1-0-LL (UK), MIROC6, MIROC-ES2L, IPSL-CM6A-LR (IPSL), EC-Earth3 (EC), MPI-ESM1-2-LR (MPI), and MRI-ESM2-0 (MRI). **(e-h)** Same as Fig. 13c-d, but including the average of the ensembles from CanESM5 (CAN), UKESM1-0-LL (UK), MIROC6, MIROC-ES2L, IPSL-CM6A-LR (IPSL), EC-Earth3 (EC), MPI-ESM1-2-LR (MPI), and MRI-ESM2-0 (MRI).

the post- and pre-overshoot climates (Fig. A2c), associated with ITCZ shifts larger than 2° (Fig. A2d), while other models like CNRM-ESM2-1 show more moderated changes, with persistent temperature changes limited to areas of the northern Atlantic

and the Southern Ocean (Fig. A2a) and ITCZ shifts limited to 1° (Fig. A2b). This may indicate a different role of heat transport changes depending on the model. For the case of SSP1-1.9, with more limited heat transport changes, the agreement between models is better both in terms of temperatures (Fig. A4a,c) and precipitation (Fig. A4b,d).

Regarding the regional extremes, the differences across models for TXx (Fig. A3a-d and Fig. A5a-d) are similar to those obtained for the global average of temperature (Fig. A1), and generally larger than the dispersion within the simulations of 330 a given model, showing a relatively small contribution of internal variability. For Rx5day the difference across models and within the simulations of a given model is more important (Fig. A3e-h and Fig. A5e-h), showing both a larger contribution of internal variability and larger differences in the modelling of regional precipitation.

## Appendix B: Selection of pre-overshoot reference period

To assess the differences between pre- and post-overshoot climates, the situation at the end of the simulations has been com-335 pared with a reference period before the overshoot. Considering that the global temperature at the end of the simulations is the same as that of 2034 for the EXT ensemble of SSP5-3.4OS and that of 2030 for the ALL ensemble of SSP1-1.9, and to have a reference period large enough to focus on the long-term variability, the period from 2020 to 2039 has been considered. However, the conclusions of this comparison may depend on the exact definition of this reference period.

Figure B1 shows a comparison between the post-overshoot situation and two alternative pre-overshoot periods (2010 to 2029 340 and 2030 to 2049). When comparing to the 2010-2029 period, the differences tend to be more positive for all the areas, and conversely more negative when comparing to the 2030-2049 period, consistent with the different level of global mean temperature for these two periods. Despite these differences, similar temperature asymmetries can be found in both comparisons. For SSP5-3.4OS, negative temperature differences are found in the northern Atlantic and large areas of northern Europe and northern Asia and positive differences are found in most of the Southern Ocean, both when considering 2010-2029 (Fig. B1a) 345 and 2030-2049 (Fig. B1c). For SSP1-1.9 the changes during the overshoot are more limited, making the comparison more sensitive to the reference period. Despite this, positive differences are consistently found in large areas of the Arctic and negative differences in mid-latitudes of the northern Atlantic (Fig. B1e,g).

For precipitation, both SSP5-3.4OS (Fig. B1b,d) and SSP1-1.9 (Fig. B1f,h) show negative differences north of the Atlantic and eastern Pacific ITCZ and positive differences to the south, both when considering 2010-2029 (Fig. B1b,f) and 2030-2049 350 (Fig. B1d,h) as reference period. This confirms that the conclusions extracted from Fig. 5 and 6 are generally robust to the definition of the pre-overshoot reference period.

*Author contributions.* PJRG contributed with conceptualization of the study, data processing, discussion of results, and writing of the paper. PDL contributed with data processing and discussion of results. RB contributed with discussion of results. MGD contributed with conceptualization of the study, discussion of results, and writing of the paper.

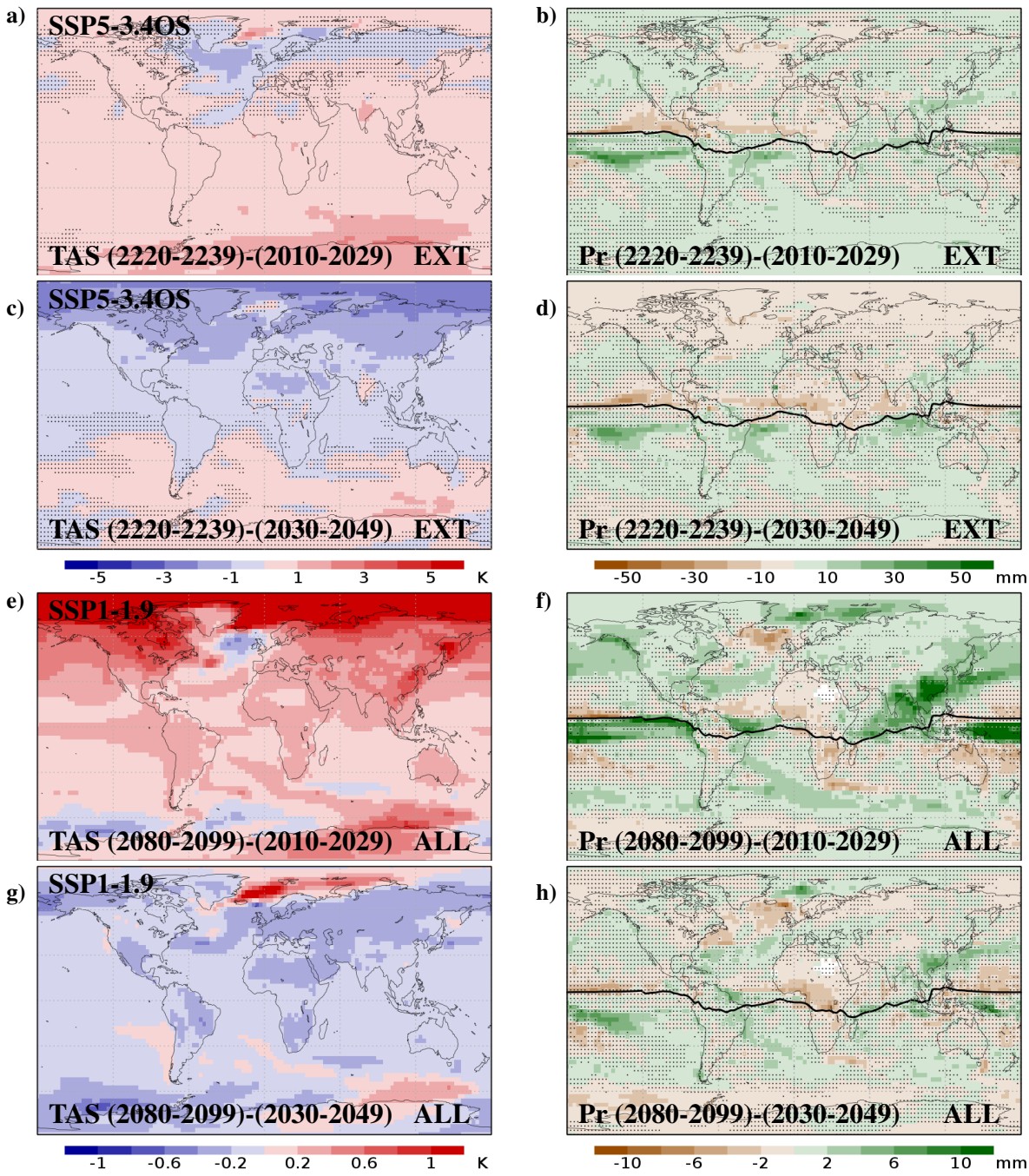

**Figure B1. (a-d)** Same as Fig. 5a-b, but considering **(a-b)** 2010-2029 and **(c-d)** 2030-2049 as pre-overshoot reference period. **(e-h)** Same as Fig. 6a-b, but considering **(e-f)** 2010-2029 and **(g-h)** 2030-2049 as pre-overshoot reference period.

*Competing interests.* The authors declare that they have no conflict of interest.

*Acknowledgements.* This research contributes to the Horizon 2020 project RESCUE (grant agreement No. 101056939) and the Spanish Ministry for Science and Innovation project PRECEDE (Grant No. EUR2022-134059). The authors are grateful to Margarida Samso-Cabre for downloading and formatting the CMIP6 input data used in the analyses.

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
