# Peer review of "Regional irreversibility of mean and extreme surface air temperature and precipitation in CMIP6 overshoot scenarios associated with interhemispheric temperature asymmetries"

_Earth System Dynamics, 2024_

## Author Comment (AC1)

**Responses to reviewer's comments for "Regional non-reversibility of mean and extreme climate conditions in CMIP6 overshoot scenarios linked to large-scale temperature asymmetries"**

We are grateful to the reviewers for their comments and suggestions, all of which have been helpful for improving the manuscript. We answer to each of the comments below, providing in gray the comments from each review and in black our responses.

**Reviewer 1:**

**R1C1**

The authors present a comprehensive analysis of regional climate signals in the overshoot scenarios SSP5-34OS and SSP1-19. The study provides valuable insights about overshoot implications and is surely of interest for a wider audience. The scope of the presented study is very similar with Pfleiderer et al. 2024 and, as the authors write in their discussion, many of the findings are consistent. Due to different methodological choices, the results of this study are still very relevant and definitely worth publishing. It would, however, be useful to discuss the methodological differences between the two studies and implications in more detail. The study is well written and I would suggest publication after some clarifications and potential changes in the framing.

As suggested by the reviewer, the methodological differences with respect to Pfleiderer et al. (2024) have been highlighted in the introduction:

*"In line with Pfleiderer et al. (2024), this work analyzes overshoot scenarios from CMIP6 (SSP5-3.4OS and SSP1-1.9) to investigate how global changes in temperature and precipitation during the overshoot are associated with regional irreversibility. Irreversibility is understood as a post-overshoot state different from the pre-overshoot state, considering pre-overshoot and post-overshoot states with the same $CO_2$ concentration and with the same global temperature. This includes then continued, partially reversed and overcompensated behaviors, as described in Pfleiderer et al. (2024). Contrary to Pfleiderer et al. (2024), who focuses on the regional reversibility up to 2100, our work includes a detailed characterisation of the stabilisation period, including also simulations of SSP5-3.4OS extending up to 2300. The analyses go also deeper into the mechanisms explaining the different regional behaviors, with an evaluation of the changes in the position of the ITCZ as a result of persistent temperature asymmetries. These analyses, including mean and extreme climates, allow not only for identification of those regions more impacted by irreversibility, but also of the mechanisms explaining different regional behaviors."*

**R1C2**

The authors have chosen to work with ensemble means which is a common approach but has some subtle implications when studying overshoot scenarios. Most importantly, different climate models reach peak warming in different years (fig A1) and in the same way, the periods when GMT stabilizes and the respective period before peak warming differs between climate models. Therefore, some of the signal seen in fig 5, 6 (...) might be influenced by considerably different GMT levels. Example: UK-model is considerably warmer in 2090 as compared to 2030 (see fig

A1.). The meaning and interpretation of "ensemble mean" becomes a bit complicated in this case as many different effects (regional hysteresis, differences in GMT trajectories, ...) will be merged in one number. It would be important to discuss these effects and if possible estimate how important they are in comparison.

Figures A2 and A4 have been added, considering the same temperature and precipitation differences as in Fig. 5 and 6 but for individual models. Discussion on the differences across the models has been included in Appendix A:

*"Regarding the spatial patterns, Fig. A2 and A4 show the same differences as in Fig. 5 and 6 but considering only some particular model simulations. In particular, the extended simulations of CNRM-ESM2-1 and MRI-ESM2-0 are presented for SSP5-3.4OS (Fig. A2) and the ensemble averages of CanESM5 and IPSL-CM6A-LR are presented for SSP1-1.9 (Fig. A4). Despite the general agreement among the models on simulating a post-overshoot climate characterised by temperature asymmetries and ITCZ shifts, for the case of SSP5-3.4OS the spatial patterns and the magnitude of the changes strongly differ across the models (Fig. A2). Some models like MRI-ESM2-0 show a strong hemispherical temperature asymmetry between the post- and pre-overshoot climates (Fig. A2c), associated with ITCZ shifts larger than 2º (Fig. A2d), while other models like CNRM-ESM2-1 show more moderate changes, with persistent temperature changes limited to areas of the northern Atlantic and the Southern Ocean (Fig. A2a) and ITCZ shifts limited to 1º (Fig. A2b). This may indicate a different role of heat transport changes depending on the model. For the case of SSP1-1.9, with more limited heat transport changes, the agreement between models is better both in terms of temperatures (Fig. A4a,c) and precipitation (Fig. A4b,d)."*

Results for the ensemble median have been included in Fig. 2, 7 and 9, together with the ensemble average. Discussion on the use of average and median has been added to the methods section:

*"The use of ensemble averages allows for a synthetic view of the results, but it may be not meaningful in case of large discrepancies across the contributing models, in particular in terms of global temperature trajectories during the overshoot and in terms of regional hysteresis. Other metrics like the ensemble median would be more robust to these effects, but they may be impacted by internal variability of individual simulations. To confirm that the ensemble average is not biased by any particular model, the ensemble median has also been computed and compared with the ensemble average. To analyze the dispersion among the models and within a single model, time series for the ensemble of each individual model providing several simulations (CanESM5 and UKESM1-0-LL for SSP5-3.4OS and CanESM5, EC-Earth3, IPSL-CM6A-LR, MIROC6, MIROC-ES2L, MPI-ESM1-2-LR, MRI-ESM2-0, and UKESM1-0-LL for SSP1-1.9) and examples of spatial patterns for some individual models (CNRM-ESM2-1 and MRI-ESM2-0 for SSP5-3.4OS, and CanESM5 and IPSL-CM6A-LR for SSP1-1.9) are also presented in Appendix A."*

Comparisons between average and median have been also added in the results section:

*"Figure 2a,c shows the global average of temperature for the experiments SSP5-3.4OS and SSP1-1.9, including both ensemble average and ensemble median"*

*"Both for SSP5-3.4OS (Fig. 7) and SSP1-1.9 (Fig. 9), similar results are obtained when using either of the ensemble average and ensemble median, confirming that the average is not biased by any individual model. A more detailed analysis on the differences between models is included in Appendix A."*

**R1C3**

Besides these technical implications of focusing on the ensemble mean, it is questionable whether ensemble mean differences are useful to inform potential risks. As Pfleiderer et al. 2024 shows (also visible in the appendix of the manuscript) the regional response differs considerably between climate models. Going into the details of model differences might be beyond the scope of this paper, but the authors could consider to show one exemplary regional difference between two models to highlight uncertainty when it comes to overshoot scenarios. In my opinion the uncertainty of climate projections in a cooling climate deserve some special attention as there is no observational data with a forced cooling trend to compare with.

Discussion about the use of ensemble means and results for individual models have been included, as per answer to R1C2.

As proposed by the reviewer, examples of temperature and precipitation differences for individual models have been included as Fig. A2 and A4.

**R1C4**

Last comment on the ensemble mean: is there a reason you don't use the ensemble median? I would find the ensemble median more appropriate as it does not give additional weight to single models with strong reactions. I would suggest to check the sensitivity to the choice of ensemble mean / ensemble median and briefly discuss this.

Discussion about the use of ensemble means and results based on ensemble median have been included, as per answer to R1C2.

**R1C5**

The authors do not mention potential effects of aerosol reductions. On a regional level, changes in aerosol emissions can considerably influence precipitation and temperature. In SSP5-34OS and SSP1-19, besides changes in GHG, aerosol emissions change and some of the regional changes around peak warming might be influenced by aerosols. The authors should at least discuss this caveat/feature of the analyzed scenarios and it's implications on the findings.

Indeed, changes in aerosol may be relevant at regional level, particularly in polar areas. The specification of anthropogenic aerosol emissions used for SSP1-1.9 and SSP5-3.4OS has been included in Fig. 2, 7, 9, A1 and A3.

The method section has been modified accordingly:

*"Even if changes in CO 2 concentration are the main contribution to the change of radiative forcing during the overshoot, changes in aerosols may also play a relevant role, particularly over the Arctic (England et al., 2021; Ren et al., 2020; DeRepentigny et al., 2022). For this reason, the emissions of aerosols for SSP5-3.4OS and SSP1-1.9 have been assessed, as provided by Feng et al. (2020)."*

The result section has been modified to highligh the link between aerosol emissions and temperature behavior in polar areas:

- *"However, SSP1-1.9 shows a persistent warming in most polar areas of the NH and cooling over the western Southern Ocean. This may be linked to the fact that even if CO 2 concentration strongly differs from SSP1-1.9 to SSP5-3.4OS, the anthropogenic aerosol emissions, more relevant in polar areas (England et al., 2021), are similar for both experiments (Fig. 2)."*

- *"The forcing conditions of SSP1-1.9 are then characterized by an opposition between high and mid latitudes rather than an opposition between NH and SH, potentially due to a delayed recovery of sea ice (Bauer et al., 2023) and a larger role of anthropogenic aerosol emissions (Fig. 2)."*

The discussion section has been modified to consider the role of aerosols:

*"For SSP1-1.9, showing less asymmetry between NH and SH and a more intense contrast between high and mid latitudes, a larger role of anthropogenic aerosol emissions and ice melting may be present, generating persistent changes in polar regions during the overshoot."*

Specific comments:

**R1C6**
Stippling in all figures: Is this a test performed on the ensemble mean? If yes, it would also be interesting to show model agreement.

Yes, the test is performed on the ensemble mean.

As discussed in the answer to R1C2, the results for some individual models have been included in Appendix A. For these results, the same test has been also performed with the individual models, showing the regions for which each individual model shows significant differences between the two periods.

However, it has been decided not to include a model agreement stippling in the figures of the main text, to avoid overloading, and considering that the reader can still refer to the Appendix A for a detailed discussion on the agreement between models.

**R1C7**
L72-73: Pfleiderer et al. 2024 focuses on exactly that question.

As discussed in the answer to R1C1, differences with respect to Pfleiderer et al. (2024) have been highlighted in the introduction.

**R1C8**

Fig 1: Why is there no brown line in the bottom of panel a & c?

The brown line represents the global average of temperature obtained with the ALL ensemble for SSP1-1.9 and the EXT ensemble for SSP5-3.4OS. This is exactly the same information that is already included with a grey solid line in Fig. 2a (for SSP5-3.4OS) and with a red solid line in Fig. 2c (for SSP1-1.9). For this reason, and to avoid redundant information in the figures, the brown line has not been included in these panels.

**R1C9**

L95-102: Just out of interest: do you have these extreme indices for all the models and runs listed in table 1? From my experience, daily data that is required for the computation of these indices is not (easily?) available for all simulations for which monthly tas and pr exists.

We computed the extreme indices for all the models and corresponding ensemble members listed in Table 1. Note that Table 1 does not contain all the runs of overshoot scenarios from CMIP6, since indeed for some of the runs the daily data was not available at the time the analyses were performed.

The extreme indices were computed with the Climpact R package (https://github.com/ARCCSS-extremes/climpact) which requires as input data the daily maximum temperature, daily minimum temperature and daily total precipitation. These input data were obtained directly from the ESGF nodes (https://esgf-node.ipsl.upmc.fr/search/cmip6-ipsl/).

**R1C10**

L103-107: I would expect that all the results would be quite sensitive to the choice of this period. Therefore, some sensitivity testing and some more discussion of the implications of different GMT levels between pre-overshoot period and the stabilization period would be helpful.

A new Appendix B has been added to the paper, including results with alternative pre-overshoot reference periods (2010 to 2029 and 2030 to 2049) in Fig. B1, and discussing the sensitivity of results to the selection of the pre-overshoot reference period:

*"To assess the differences between pre- and post-overshoot climates, the situation at the end of the simulations has been compared with a reference period before the overshoot. Considering that the global temperature at the end of the simulations is the same as that of 2034 for the EXT ensemble of SSP5-3.4OS and that of 2030 for the ALL ensemble of SSP1-1.9, and to have a reference period large enough to focus on the long-term variability, the period from 2020 to 2039 has been considered. However, the conclusions of this comparison may depend on the exact definition of this reference period.*

*Figure B1 shows a comparison between the post-overshoot situation and two alternative pre-overshoot periods (2010 to 2029 and 2030 to 2049). When comparing to the 2010-2029 period, the differences tend to be more positive for all the areas, and conversely more negative when comparing to the 2030-2049 period, consistent with the different level of global mean temperature for these two periods. Despite these differences, similar temperature asymmetries can be found in both comparisons. For SSP5-3.4OS, negative temperature differences are found in the northern Atlantic and large areas of northern Europe and northern Asia and positive differences are found in most of the Southern Ocean, both when considering 2010-2029 (Fig. B1a) and 2030-2049 (Fig. B1c). For SSP1-1.9 the changes during the overshoot are more limited, making the comparison more sensitive to the
reference period. Despite this, positive differences are consistently found in large areas of the Arctic and negative differences in mid-latitudes of the northern Atlantic (Fig. B1e,g).*

*For precipitation, both SSP5-3.4OS (Fig. B1b,d) and SSP1-1.9 (Fig. B1f,h) show negative differences north of the Atlantic and eastern Pacific ITCZ and positive differences to the south, both when considering 2010-2029 (Fig. B1b,f) and 2030-2049 (Fig. B1d,h) as reference period. This confirms that the conclusions extracted from Fig. 5 and 6 are generally robust to the definition of the pre-overshoot reference period"*

Reference to the Appendix has been also added in the main text:

*"To confirm the suitability of this reference period, results obtained with alternative pre-overshoot periods from 2010 to 2029 and from 2030 to 2049 have been also included in Appendix B."*

**R1C11**
L141-149: For the comparison of the two scenarios, it would be useful to merge fig. 2 and fig. 3. Since the GMT trajectory differs between the two scenarios, it is a bit unclear what conclusions can be drawn from the comparison of the two scenarios when fixed periods are used (as you also show in fig 4).

Indeed, the comparison of the two scenarios is mostly focused on the spatial patterns rather than on the temporal evolution, since the temporal evolution is primarily driven by the GMT trajectory, which differs between the two scenarios. As suggested by the reviewer, Fig. 2 and Fig. 3 have been merged in the new Fig. 2.

**R1C12**
Fig 4: "obtained as the year after the maximum in which temperature reaches the same value as in the period 2290-2300" -> what is the tolerance for the temperature differences? And are you comparing 20-year periods with the 10-year period at the end (2290-2300)? I think that for this comparison, the period should have the same length. Similar question for e) and f): are you comparing single years, or 20-year periods?

For the stabilisation year in Fig. 4a and for the year with the same temperature as in 2034 and 2015 in Fig. 4e-f, individual years are considered. This is because the data is already filtered with a

moving average of 10 years, as stated in the methods section (*"temporal evolutions filtered with a 10 year moving average"*), so when comparing the individual years with a 10-year period at the end (2290-2300) we are indeed comparing periods with the same length.

Periods of 20 years are considered for Fig. 5, 6, 7 and 9, in which they are compared with a reference period of 20 years before the overshoot (2020-2039), but not for Fig. 4. More discussion on the selection of this reference period is included in the answer to R1C10.

**R1C13**
L175-177: How do you interpret that the ITCZ shift is lower in SSP1-19 but precipitation differences are higher?

The precipitation differences for SSP1-1.9 (Fig. 6) are indeed lower than the precipitation differences for SSP5-3.4OS (Fig. 5). Please, note the different scale used for the two figures (50 mm vs 10 mm).

**R1C14**
L193-195: Does the word "being" belong here?

It has been replaced by *"the same before and after the overshoot, with persistent changes mostly produced around the maximum"*.

**R1C15**
L199-201: Although the slope looks similar, there is a different TXx - GMT relationship after the overshoot. At the same GMT level after the overshoot one would expect a lower TXx value in EN, right?

Yes, indeed. The slope is similar, but it remains at a lower TXx than before the overshoot for EN and at a higher TXx than before the overshoot for ES. Sentence has been modified to clarify this:

*"The scaling of regional extremes with the global mean temperature is recovered afterwards, and starting from 2100 TXx decreases linearly with respect to the global average of temperature, both for EN and EH (Fig. 7f), with a slope similar to that of the increasing phase between 2000 and 2060, but vertically shifted keeping the TXx differences cumulated during the transition period."*

**R1C16**
L251-253: Again, I'm not sure if I would agree. Isn't this statement contradicting L234-236?:

As per answer to R1C15, this refers to the slope of the GMT-TXx relationship, and not to the offset. The same slope as in the pre-overshoot climate is recovered, but there is a remaining offset cumulated during the transition phase. Sentence has been modified to clarify this:

*"Changes in the relationship between the regional climate conditions and the global mean temperature mainly take place during the transition period around the global temperature*

*maximum (from 2060 to 2080 for SSP5-3.4OS and from 2040 to 2060 for SSP1-1.9). Afterwards, the relationship between global mean temperatures and regional extremes recovers a similar slope to that of the pre-overshoot period, but with an offset cumulated during the transition phase."*

**Reviewer 2:**

The paper analyzes the response of surface air temperature and precipitation under two CMIP6 overshoot scenarios. The authors focus on mean and extreme changes of them. It would be a very interesting contribution to understanding how the climate system changes under the overshooting scenarios. I support publication, but some parts should be revised to publish in ESD. Please find the comments below:

Comments:

**R2C1**
- Title: The current title is vague and does not effectively convey the actual content of the result. First of all, 'asymmetries' can have dual meanings without knowing the actual content of the paper. It can mean either asymmetry between global warming and cooling periods or between the Northern and Southern Hemisphere. Therefore, somehow, the title must be changed. The '..linked to large-scale temperature asymmetries' part emphasizes the mechanistic part of the result, but I suggest removing it because the paper does not deeply dive into the mechanistic analysis (roughly done though) and not the major novel part. My suggestions are, for example:

'Hysteresis of mean and extreme surface temperature and precipitation in the CMIP6 overshoot scenario', 'Response of mean and extreme surface temperature and precipitation in the CMIP6 overshoot scenario', 'Regional irreversibility of mean and extreme surface temperature and precipitation in the CMIP6 overshoot scenario'.

As suggested by the reviewer, the title has been modified to:

- Clarify that it refers to interhemispheric temperature asymmetries and not to asymmetries between global warming and cooling phases.

- Specify that the analyses are performed for mean and extreme surface air temperature and precipitation.

The reference to temperature asymmetries has not been removed. The authors consider that this reference is still meaningful, since it clarifies the exact mechanisms that are discussed in the paper.

According to this, the new title is: *"Regional irreversibility of mean and extreme surface air temperature and precipitation in CMIP6 overshoot scenarios associated with interhemispheric temperature asymmetries"*.

- Differentiate hysteresis and reversibility & Clear definition of reversibility: The author clearly defines what hysteresis is in the introduction part of the paper. I appreciate it. However, the irreversibility should be clearly defined as quantitative as possible. For example, how the reference period is set when we say the climate variable is irreversible? Putting a clear definition of it would be very helpful.

A detailed discussion on the selection of reference period is included in the method section:

*"The situation after stabilization has been compared with the situation before the overshoot with the same $CO_2$ concentration, as provided by Meinshausen et al. (2020), reached in 2015 both for SSP5-3.4OS and SSP1-1.9. It has been also compared with the situation with the same global temperature, reached in 2034 for SSP5-3.4OS and in 2030 for SSP1-1.9. Considering these dates and to use a reference period large enough to focus on the long-term variability, the period from 2020 to 2039 has been considered as pre-overshoot reference period for most of the analyses."*

As per answer to R1C10, a new Appendix B has been added to the paper, including results with alternative pre-overshoot reference periods (2010 to 2029 and 2030 to 2049) in Fig. B1, and discussing the sensitivity of results to the selection of the pre-overshoot reference period.

Introduction has been modified to define more in detail irreversibility:

*"In line with Pfleiderer et al. (2024), this work analyzes overshoot scenarios from CMIP6 (SSP5-3.4OS and SSP1-1.9) to investigate how global changes in temperature and precipitation during the overshoot are associated with regional irreversibility. Irreversibility is understood as a post-overshoot state different from the pre-overshoot state, considering pre-overshoot and post-overshoot states with the same CO2 concentration and with the same global temperature. This includes then continued, partially reversed and overcompensated behaviors, as described in Pfleiderer et al. (2024)."*

**R2C3**

- The term 'non-reversibility': The author uses 'non-reversibility' not 'irreversibility' throughout the paper. English-wise sense both make sense, but the later one is much more used in the literature and much more standard term than the first one. I am wondering why the authors decided to choose this unpopular term. Is there any specific reason for it? I suggest changing the term to 'irreversibility' if there is no specific reason for it.

There is no specific reason to use "non-reversibility". As suggested, all the occurrences of "non-reversibility" have been replaced by "irreversibility".

**R2C4**

- There are multiple simulations within the same model with the same model forcing scenario. For example, Table 1 shows that CanESM5 has 50 simulations for the SSP1-1.9 scenario. I assume that they're an ensemble with different initial conditions, but not sure and puzzling. Please clarify this

information. The paper must include which ensemble member is used for the analysis.

Yes, the simulations of the same model and experiment are an ensemble with the same forcing conditions and different initial conditions.

This has been clarified in the text:

*"For that, the simulations in Table 1 have been considered. For some of the models, several simulations with the same forcing specifications and different initial conditions are considered."*

Table 1 has been modified to include the exact ensemble members used for the analysis.

**R2C5**
- Additional analysis for ITCZ and AMOC: An additional time series plot for ITCZ position (i.e., plot for ITCZ position-time) would be very helpful for understanding the results. I also suggest performing additional analysis for AMOC strength which has a very important role in shaping hysteresis.

Fig. 5c and 6c have been added, including time series of the average position of the Atlantic ITCZ (70ºW - 25ºE), computed with the precipitation centroid, and time series of the average southwards Ocean Heat Transport (OHT) in the Atlantic basin, representative of the AMOC strength.

Discussion has been also added in the results section:

*"In the Atlantic basin, the changes in the ITCZ position can be associated with changes in the ocean heat transport (Fig. 5c), linked to a decline of the AMOC."*

**R2C6**
- To calculate the extreme indices (e.g., Rx1day), a baseline climatology value should be chosen. Under the changing climate, the way of defining this baseline climatology is very important (this is a reference for the Marine Heatwave case, but relevant to this case: https://www.nature.com/articles/d41586-023-00924-2). How did the author set the baseline climatology value for the extreme indices? This is very important for interpreting results, but not clearly explained.

The reference period that the reviewer refers to is relevant for extremes indices defined as exceedence of a percentile threshold. While in the context of climate impacts, a moving reference period may be useful to account for potential adaptation, the analysis of purely physical extremes in a forced climate needs a fixed reference period to ensure consistency/comparability of the extremes at different forcing levels.

We would further like to point out that most of the extremes presented in this study (incl. Rx1day) are not calculated relative to a percentile threshold (only TX90p/TN10p are). That rationale for presenting Rx1day and Rx5day changes as percentages compared to the average of the first 20

years is primarily to remove the effect of large inter-model differences in the absolute precipitation amounts (in mm) and instead consider changes in % (which also helps with physical interpretations related to e.g. the Clausius-Clapeyron relationship).

The definition of indices considered in this work is the one from Zhang et al. (2011). In particular, Rx1day is defined as the monthly maximum 1-day precipitation, independently of the climatology. For Fig. 11, 12, 13 and 14, a reference period from 1861 to 1880 is considered to express the precipitation in percentage change rather than in mm, but this reference period is only used for changing the units of precipitation and not for the computation of the extreme indices. This is detailed in the caption of the figures: *"Rx5day and Rx1day are expressed in percentage of variation with respect to 1861-1880."*

**R2C7**
- For the spatial pattern analysis (e.g., Figure 2 and 3), the author chose to do analysis within the continuously moving time frame with a 20-years period length, and not compared with the anomaly from the fixed preindustrial period as shown in Fig. 1. This hinders consistent comparison between the spatially averaged results and spatially-resolved results. Is there any specific reason to take such an analysis scheme?

The selection of a moving time frame of 20 years for Fig. 3 is intended to compare the spatial patterns during the global temperature increase (Fig. 3a-b) and decrease (Fig. 3b-c) phases. A comparison with the pre-industrial period would mostly show the spatial patterns of the increase phase, limiting then this comparison.

**R2C8**
- Abstract & Conclusion: The paper contains a lot of results with a large number of figures. Therefore, it is important to concisely & systematically organize the result in the paper, in particular in the abstract and conclusion section The current abstract and conclusion part requires heavy revision in this sense. I suggest reframing the abstract and conclusion part following the structure below:

'How the global mean/extreme surface air temperature and precipitation change'
'How do the regional mean/extreme surface air temperature and precipitation change, in particular, which part of the region shows the strong and weak reversibility under the overshooting scenario?'
'Mechanistic insights into why such changes happen – role of ITCZ (and AMOC; optionally).'
'Discussion on decoupling between regional and global average response'.

Abstract has been rewritten to follow the proposed structure:

'How the global mean/extreme surface air temperature and precipitation change'

*"These analyses show that in scenarios with strong forcing changes like SSP5-3.4OS, the post-overshoot state is characterised by a temperature asymmetry between Northern and Southern Hemisphere, associated with shifts of the Intertropical Convergence Zone (ITCZ). In scenarios with*

*lower forcing changes like SSP1-1.9, this hemispheric asymmetry is more limited and temperature changes in polar areas are more prominent."*

'How do the regional mean/extreme surface air temperature and precipitation change, in particular, which part of the region shows the strong and weak reversibility under the overshooting scenario?'

*"These large scale changes have an impact on regional climates, such as for temperature extremes in extratropical regions and for precipitation extremes in tropical regions around the ITCZ."*

'Mechanistic insights into why such changes happen – role of ITCZ (and AMOC; optionally).'

*"Differences between pre- and post-overshoot states may be associated with persistent changes in the heat transport and with a different thermal inertia depending on the region, leading regionally to a different timing of the temperature maximum. Other factors like changes in aerosol emissions and ice melting may be also important, particularly for polar areas."*

'Discussion on decoupling between regional and global average response'.

*"Results show that irreversibility of temperature and precipitation extremes is mainly caused by the transitions around the global temperature maximum, when a decoupling between regional extremes and global temperature generates persistent changes at regional level."*

In a similar way, conclusions have been reorganised to follow the proposed structure:

'How the global mean/extreme surface air temperature and precipitation change'

*"For SSP5-3.4OS, the situation after the overshoot is characterized by a colder NH and a warmer SH, associated with a southward shift of the ITCZ, in line with the results in idealized experiments (Kug et al., 2022). The analysis of SSP1-1.9 is limited by the period covered by simulations. Even if the maximum of temperature for this experiment is reached for most regions before 2050, the climate is not fully stabilized by 2100, when the simulations end. Even with that, the analysis of the final state shows significant differences with respect to the situation before the overshoot, with higher temperatures for polar regions of the NH and for certain areas of the Southern Ocean, and with ITCZ shifts, to the south over the Pacific and Atlantic basin and to the north over the Indian basin."*

'How do the regional mean/extreme surface air temperature and precipitation change, in particular, which part of the region shows the strong and weak reversibility under the overshooting scenario?'

*"Changes in temperature and precipitation during the overshoot may explain relevant changes and hysteresis in regional extremes. Warmest regional temperatures after overshoot exceed those obtained at the same global average temperature before the overshoot for most tropical and extratropical regions of the SH in SSP5-3.4OS and for high-latitude regions both of the NH and SH in SSP1-1.9. This is consistent with, and can explain, the partially reversed behavior found by*

*Pfleiderer et al. (2024) in 2100 for the TXx of RAR, NEU, GIC, NEN, NZ, and SSA (for the region definitions see Fig. 1). The persistent changes are even larger for the coldest temperatures, showing a significant decline in many continental regions of the NH both for SSP5-3.4OS and SSP1-1.9. This was also found by Pfleiderer et al. (2024) for the TNn of WCA, SAH, and TIB, with a partially reversed behavior in 2100, but not so clearly for other regions like MED, WCE, and EEU, where the stabilization is reached after 2100 (Fig. 4d). Despite the minor role of hysteresis found by Walton and Huntingford (2024) for the regional precipitation of tropical areas, a relevant role is found in regions around the ITCZ. Precipitation extremes for these regions are impacted by ITCZ shifts, with*
*both experiments showing a decline in the intensity of extreme precipitation in regions to the north of the ITCZ, like Western and Central Africa, in line with the overcompensated behavior found by Pfleiderer et al. (2024) for these regions."*

'Mechanistic insights into why such changes happen – role of ITCZ (and AMOC; optionally).'

*"For SSP5-3.4OS, the fact that the maximum of regional temperatures is reached before 2070 for most continental areas and after 2090 for the Southern Ocean suggests a relevant role of the inertia of the ocean, experiencing warming and cooling phases delayed compared to those of continental areas. However, other mechanisms like changes in the AMOC (Moreno-Chamarro et al., 2020) or changes in sea ice (Li et al., 2020) may also contribute. For SSP1-1.9, showing less asymmetry between NH and SH and a more intense contrast between high and mid latitudes, a larger role of anthropogenic aerosol emissions and ice melting may be present, generating persistent changes in polar regions during the overshoot."*

'Discussion on decoupling between regional and global average response'.

*" Changes in the relationship between the regional climate conditions and the global mean temperature mainly take place during the transition period around the global temperature maximum (from 2060 to 2080 for SSP5-3.4OS and from 2040 to 2060 for SSP1-1.9). Afterwards, the relationship between global mean temperatures and regional extremes recovers a similar slope to that of the pre-overshoot period, but with an offset cumulated during the transition phase. The evolution of regional extremes is mostly coupled to the evolution of the global temperatures during the periods of increasing and decreasing global temperature, but it is decoupled during the transition period around the global maximum depending on the timing of regional maximum temperatures, generating region-dependent irreversibilities."*

**R2C9**

- More systematic analysis: The paper mainly merits showing the response of two key climate variables, surface temperature and precipitation under the overshooting scenarios. Mechanistic analysis part is not as novel as the main results. It is really good and would be very helpful for the community and also for the general public. However, the current analysis of the paper lacks the systematic. The ideal arrangement of the paper is:

Changes in mean surface temperature (global-average & regional pattern)

Providing this information systematically would be 'very' beneficial. Some parts are missing and require additional analysis. Also, the paper requires some rearrangement of the result sections (for example, figure B1, B2, B3 are very useful and can be placed in the main figure). I assume that the author intended to emphasize some specific point of their main results, but plainly providing the above information is already okay and novel. I strongly suggest considering this suggestion.

As suggested, Appendix B has been removed and the content has been included in the main text (Fig. 1, 8, 10, 12 and 14).

The text has been also reorganised, with the following structure:

- Changes in mean surface air temperature
- Changes in mean precipitation
- Changes in extreme surface air temperature
- Changes in extreme precipitation

This new structure contains the same sections as suggested by the reviewer, but it has been decided to present first all the results for mean variables (temperature and precipitation) and later all the results for extremes (temperature and precipitation), mainly because this structure allows for an easier interpretation of the mean precipitation results (linked to mean temperature and not to extreme temperature).

Figures have been adapted accordingly, separating results in:

- Changes in mean surface air temperature: Fig. 2-4
- Changes in mean precipitation: Fig. 5-6
- Changes in extreme surface air temperature: Fig. 7-10
- Changes in extreme precipitation: Fig. 11-14